# Rethinking Symbolic Regression Datasets and Benchmarks for Scientific Discovery

## Abstract

This paper revisits datasets and evaluation criteria for Symbolic Regression, a task of recovering mathematical expressions from given data, specifically focused on its potential for scientific discovery. Focused on a set of formulas used in the existing datasets based on Feynman Lectures on Physics, we recreate 120 datasets to discuss the performance of symbolic regression for scientific discovery (SRSD). For each of the 120 SRSD datasets, we carefully review the properties of the formula and its variables to design reasonably realistic sampling ranges of values so that our new SRSD datasets can be used for evaluating the potential of SRSD such as whether or not an SR method can (re)discover physical laws from such datasets. As an evaluation metric, we also propose to use normalized edit distances between a predicted equation and the ground-truth equation trees. While existing metrics are either binary or errors between the target values and an SR model's predicted values for a given input, normalized edit distances evaluate a sort of similarity between the ground-truth and predicted equation trees. We have conducted experiments on our new SRSD datasets using five state-of-the-art SR methods in SRBench and a simple baseline based on a recent Transformer architecture. The results show that we provide a more realistic performance evaluation and open up a new machine learning-based approach for scientific discovery. We provide our datasets and code as part of the supplementary material.

## 1 Introduction

Recent advances in machine learning (ML), especially deep learning (DL), have led to the proposal of many methods that can reproduce the given data and make appropriate inferences on new inputs. Such methods are, however, often black-box, which makes it difficult for humans to understand how they made predictions for given inputs. This property will be more critical especially when non-ML experts apply ML to problems in their research domains such as physics and chemistry.

Symbolic regression (SR) is the task of producing a mathematical expression (symbolic expression) that fits a given dataset. SR has been studied in the genetic programming (GP) community (Hoai et al., 2002; Keijzer, 2003; Koza & Poli, 2005; Johnson, 2009; Uy et al., 2011; Orzechowski et al., 2018), and DL-based SR has been attracting more attention from the ML/DL community (Petersen et al., 2020; Landajuela et al., 2021; Biggio et al., 2021; Valipour et al., 2021; La Cava et al., 2021; Kamienny et al., 2022). Because of its interpretability, various scientific communities apply SR to advance research in their scientific fields *e.g.*, Physics (Wu & Tegmark, 2019; Udrescu & Tegmark, 2020; Udrescu et al., 2020; Kim et al., 2020; Cranmer et al., 2020; Liu & Tegmark, 2021; Liu et al., 2021b), Applied Mechanics (Huang et al., 2021), Climatology (Abdellaoui & Mehrkanoon, 2021), Materials (Sun et al., 2019; Wang et al., 2019; Weng et al., 2020; Loftis et al., 2020), and Chemistry (Batra et al., 2020).

Given that SR has been studied in various communities, La Cava et al. (2021) propose SRBench, a unified benchmark framework for symbolic regression methods. In the benchmark study, they combine the Feynman Symbolic Regression Database (FSRD) (Udrescu & Tegmark, 2020) and the ODE-Strogatz repository (Strogatz, 2018) to compare a number of SR methods, using a large-scale heterogeneous computing cluster.[1]

---

[1]They used hosts with 24-28 core Intel(R) Xeon(R) CPU E5-2690 v4 2.60GHz processors and 250GB RAM.

To discuss the potential of symbolic regression for scientific discovery (SRSD), there still remain some issues to be addressed: oversimplified datasets and lack of evaluation metric towards SRSD. For symbolic regression tasks, existing datasets consist of values sampled from limited domains such as in range of 1 to 5, and there are no large-scale datasets with reasonably realistic values that capture the properties of the formula and its variables. Thus, it is difficult to discuss the potential of symbolic regression for scientific discovery with such existing datasets. For instance, the FSRD consists of 120 formulas selected mostly from Feynman Lectures Series[2] (Feynman et al., 1963a;b;c) and are core benchmark datasets used in SRBench (La Cava et al., 2021). While the formulas indicate physical laws, variables and constants used in each dataset have no physical meanings since the datasets are not designed to discover the physical laws from the observed data in the real world. (See Section 3.1.)

Moreover, there is a lack of appropriate metrics to evaluate these methods for SRSD. An intuitive approach would be to measure the prediction error or correlation between the predicted values and the target values in the test data, as in standard regression problems. However, low prediction errors could be achieved even by complex models that differ from the original law. In addition, SRBench (La Cava et al., 2021) presents the percentage of agreement between the target and the estimated equations. But in such cases, both 1) equations that do not match at all and 2) that differ by only one term[3] are equally treated as incorrect. As a result, it is considered as a coarse-resolution evaluation method for accuracy in SRSD, which still needs more discussion towards real-world applications. A key feature of SR is its interpretability, and some studies (Udrescu et al., 2020; La Cava et al., 2021) use complexity of the predicted expression as an evaluation metric (the simpler the better). However, it is based on a big assumption that a simpler expression may be more likely to be a hidden law in the data (scientific discovery such as physics law), which may not be true for SRSD. Therefore, there are no single evaluation metrics proposed to take into account both the interpretability and how close to the true expression the estimated expression is.

To address these issues, we propose new SRSD datasets, introduce a new evaluation method, and conduct benchmark experiments using representative SR methods and a new Transformer-based SR baseline. We carefully review and design annotation policies for the new datasets, considering the properties of the physics formulas. Besides, given that a formula can be represented as a tree structure, we introduce a normalized edit distance on the tree structure to allow quantitative evaluation of predicted formulas that do not perfectly match the true formulas. Using the proposed SRSD datasets and evaluation metric, we perform benchmark experiments with a set of SR baselines and find that there is still significant room for improvements in terms of the new evaluation metric.

## 2 RELATED STUDIES

In this section, we briefly introduce related studies focused on 1) symbolic regression for scientific discovery and 2) symbolic regression dataset and evaluation.

### 2.1 SRSD: SYMBOLIC REGRESSION FOR SCIENTIFIC DISCOVERY

A pioneer study on symbolic regression for scientific discovery is conducted by Schmidt & Lipson (2009), who propose a data-driven scientific discovery method. They collect data from standard experimental systems like those used in undergrad physics education: an air-track oscillator and a double pendulum. Their proposed algorithm detects different types of laws from the data such as position manifolds, energy laws, and equations of motion and sum of forces laws.

Following the study, data-driven scientific discovery has been attracting attention from research communities and been applied to various domains such as Physics (Wu & Tegmark, 2019; Udrescu & Tegmark, 2020; Udrescu et al., 2020; Kim et al., 2020; Cranmer et al., 2020; Liu & Tegmark, 2021; Liu et al., 2021b), Applied Mechanics (Huang et al., 2021), Climatology (Abdellaoui & Mehrkanoon, 2021), Materials (Sun et al., 2019; Wang et al., 2019; Weng et al., 2020; Loftis et al., 2020), and Chemistry (Batra et al., 2020).

---

[2]Udrescu & Tegmark (2020) extract 20 of the 120 equations as "bonus" from other seminal books (Goldstein et al., 2002; Jackson, 1999; Weinberg, 1972; Schwartz, 2014).

[3]If those differ by a constant or scalar, SRBench treats the estimated equation as correct for solution rate.

These studies leverage symbolic regression in different fields. While general symbolic regression tasks use synthetic datasets with limited sampling domains for benchmarks, many of the SRSD studies collect data from the real world and discuss how we could leverage symbolic regression toward scientific discovery.

While SRSD tasks share the same input-output interface with general symbolic regression (SR) tasks (*i.e.*, input: dataset, output: symbolic expression), we differentiate SRSD tasks in this study from general SR tasks by whether or not the datasets including true symbolic expressions are created with reasonably realistic assumptions for scientific discovery such as meaning of true symbolic expressions (whether or not they have physical meanings) and sampling domains for input variables.

## 2.2 DATASET AND EVALUATION

For symbolic regression methods, there exist several benchmark datasets and empirical studies. The Feynman Symbolic Regression Database (Udrescu & Tegmark, 2020) is one of the largest symbolic regression datasets, which consists of 100 physics-inspired equations based on Feynman Lectures on Physics (Feynman et al., 1963a;b;c). By randomly sampling from small ranges of value, they generate the corresponding tabular datasets for the 100 equations. Inspired by Hoai et al. (2002); Keijzer (2003); Johnson (2009), Uy et al. (2011) suggest 10 different real-valued symbolic regression problems (functions) and create the corresponding dataset (*a.k.a.* Nguyen dataset). The suggested functions consist of either 1 or 2 variables *e.g.*, $f(x) = x^6 + x^5 + x^4 + x^3 + x^2 + x$ and $f(x, y) = \sin(x) + \sin(y^2)$. They generate each dataset by randomly sampling 20 - 100 data points.

La Cava et al. (2021) design a symbolic regression benchmark, named SRBench, and conduct a comprehensive benchmark experiment, using existing symbolic regression datasets such as the Feynman Symbolic Regression Database (Udrescu & Tegmark, 2020) and ODE-Strogatz repository (La Cava et al., 2016). In SRBench, symbolic regression methods are assessed by 1) an error metric based on squared error between target and estimated values, and 2) solution rate that shows a percentage of the estimated symbolic regression models that match the true models (equations).

However, these datasets and evaluations are not necessarily designed to discuss symbolic regression for scientific discovery. In Sections 3.1 and 4.1, we further describe potential issues in prior studies.

## 3 DATASETS

In this section, we summarize issues we found in the existing symbolic regression datasets, and then propose new datasets to address them towards symbolic regression for scientific discovery (SRSD).

## 3.1 ISSUES IN EXISTING DATASETS

As introduced in Section 2.2, there are many symbolic regression datasets. However, we consider that novel datasets are required to discuss SRSD for the following reasons:

1. **No physical meaning:** Many of the existing symbolic regression datasets (Hoai et al., 2002; Keijzer, 2003; Johnson, 2009; Uy et al., 2011) are not necessarily physics-inspired, but instead randomly generated *e.g.*, $f(x) = \log(x)$, $f(x, y) = xy + \sin((x - 1)(y - 1))$. To discuss the potential of symbolic regression for scientific discovery, we need to further elaborate datasets and evaluation metrics, considering how we would leverage symbolic regression in practice.

2. **Oversimplified sampling process:** While some of the datasets are physics-inspired such as the Feynman Symbolic Regression Database (FSRD) (Udrescu & Tegmark, 2020) and ODE-Strogatz repository (La Cava et al., 2016), their sampling strategies are very simplified. Specifically, the strategies do not distinguish between constants and variables *e.g.*, speed of light[4] is treated as a variable and randomly sampled in range of 1 to 5. Besides, most of the sampling domains are far from values we could observe in the real world *e.g*, II.4.23 in Table S1 (the vacuum permittivity values are sampled from range of 1 to 5). When sampled ranges of the distributions are narrow, we cannot distinguish Lorentz transformation from Galilean transformation *e.g* I.15.10 and I.16.6 in Table S3, I.48.2 in Table S5, I.15.3t, I.15.3x, and I.34.14 in Table S7, or the black body

---

[4]We treat speed of light as a constant ($2.998 \times 10^8 \text{m/s}$) in this study.

radiation can be misestimated to Stephan-Boltzmann law or the Wien displacement law *e.g.* I.41.16 in Table S8.

3. **Duplicate equations:** Due to the two issues above, many of the equations in existing datasets turn out to be duplicate. *e.g.*, as shown in Table 1, $F = \mu N_n$ (I.12.1) and $F = q_2 E$ (I.12.5) in the *original* Feynman Symbolic Regression Database are considered identical since both the equations are multiplicative and consists of two variables, and their sampling domains (Distributions in Table 1) are exactly the same. For instance, approximately 25% of the symbolic regression problems in the *original* FSRD have 1 - 5 duplicates in that regard.

4. **Incorrect/Inappropriate formulas:** The Feynman Symbolic Regression Database (Udrescu & Tegmark, 2020) treat every variables as float whereas they should be integer to be physically meaningful. For example, the number of phase difference in Bragg's law should be integer but sampled as real number (I.30.5 in Table S1). Furthermore, they don't even give special treatment of angle variables (I.18.12, I.18.16, and I.26.2 in Table 1). Physically some variables can be negative whereas the *original* Feynman Symbolic Regression Database (Udrescu & Tegmark, 2020) only samples positive values (*e.g.* I.8.14 and I.11.19 in Table S3). We also avoid using *arcsin/arccos* in the equations since the use of *arcsin/arccos* in the Feynman Symbolic Regression Database (Udrescu & Tegmark, 2020) just to obtain angle variable is not experimentally meaningful (I.26.2 in Table 1, I.30.5 in Table S1, and B10 in Table S11). Equations using *arcsin* and *arccos* in the original annotation are I.26.2 (Snell's law), I.30.5 (Bragg's law), and B10 (Relativistic aberration). These are all describing physical phenomena related to two angles, and it is an unnatural deformation to describe only one of them with an inverse function. Additionally, inverse function use implicitly limits the range of angles, but there is no such limitation in the actual physical phenomena.

## 3.2 PROPOSED SRSD DATASETS

We address the issues in existing datasets above by proposing new SRSD datasets based on the equations used in the FSRD (Udrescu & Tegmark, 2020). *i.e.*, Section 3.1 summarizes the differences between the FSRD and our SRSD datasets. Our annotation policy is carefully designed to simulate typical physics experiments so that the SRSD datasets can engage studies on symbolic regression for scientific discovery in the research community.

### 3.2.1 ANNOTATION POLICY

We thoroughly revised the sampling range for each variable from the annotations in the FSRD (Udrescu & Tegmark, 2020). First, we reviewed the properties of each variable and treated physical constants (*e.g.*, light speed, gravitational constant) as constants while such constants are treated as variables in the original FSRD datasets. Next, variable ranges were defined to correspond to each typical physics experiment to confirm the physical phenomenon for each equation. We also used (of Japan, 2022) as a reference. In cases where a specific experiment is difficult to be assumed, ranges were set within which the corresponding physical phenomenon can be seen. Generally, the ranges are set to be sampled on log scales within their orders as $10^2$ in order to take both large and small changes in value as the order changes. Variables such as angles, for which a linear distribution is expected are set to be sampled uniformly. In addition, variables that take a specific sign were set to be sampled within that range. Tables 1 and S1 – S11 show the detailed comparisons between the original FSRD and our proposed SRSD datasets.

### 3.2.2 COMPLEXITY-AWARE DATASET CATEGORIES

While the proposed datasets consist of 120 different problems, there will be non-trivial training cost required to train a symbolic regression model for all the problems individually (La Cava et al., 2021) *i.e.*, there will be 120 separate training sessions to assess the symbolic regression approach. To allow more flexibility in assessing symbolic regression models for scientific discovery, we define three clusters of the proposed datasets based on their complexity: *Easy*, *Medium*, and *Hard* sets, which consist of 30, 40, and 50 different problems respectively.

We define the complexity of problem, using the number of operations to represent the true equation tree and range of the sampling domains. The former measures how many mathematical operations compose the true equation such as *add*, *mul*, *pow*, *exp*, and *log* operations (see Fig. 2). The latter

Table 1: Easy set of our proposed datasets (part 1). C: Constant, V: Variable, F: Float, I: Integer, P: Positive, N: Negative, NN: Non-Negative, $\mathcal{U}$: Uniform distribution, $\mathcal{U}_{\log}$: Log-Uniform distribution.

| Eq. ID | Formula | | Symbols | Properties | | Distributions | |
|---|---|---|---|---|---|---|---|
| | | | | Original | Ours | Original | Ours |
| I.12.1 | $F = \mu N_{\mathrm{n}}$ | $F$ | Force of friction | V, F | V, F, P | N/A | N/A |
| | | $\mu$ | Coefficient of friction | V, F | V, F, P | $\mathcal{U}(1,5)$ | $\mathcal{U}_{\log}(10^{-2}, 10^0)$ |
| | | $N_{\mathrm{n}}$ | Normal force | V, F | V, F, P | $\mathcal{U}(1,5)$ | $\mathcal{U}_{\log}(10^{-2}, 10^0)$ |
| I.12.4 | $E = \frac{q_1}{4\pi\epsilon r^2}$ | $E$ | Magnitude of electric field | V, F | V, F | N/A | N/A |
| | | $q_1$ | Electric charge | V, F | V, F | $\mathcal{U}(1,5)$ | $\mathcal{U}_{\log}(10^{-3}, 10^{-1})$ |
| | | $r$ | Distance | V, F | V, F, P | $\mathcal{U}(1,5)$ | $\mathcal{U}_{\log}(10^{-2}, 10^0)$ |
| | | $\epsilon$ | Vacuum permittivity | V, F | C, F, P | $\mathcal{U}(1,5)$ | $8.854 \times 10^{-12}$ |
| I.12.5 | $F = q_2 E$ | $F$ | Force | V, F | V, F | N/A | N/A |
| | | $q_2$ | Electric charge | V, F | V, F | $\mathcal{U}(1,5)$ | $\mathcal{U}_{\log}(10^{-3}, 10^{-1})$ |
| | | $E$ | Electric field | V, F | V, F | $\mathcal{U}(1,5)$ | $\mathcal{U}_{\log}(10^1, 10^3)$ |
| I.14.3 | $U = mgz$ | $U$ | Potential energy | V, F | V, F, P | N/A | N/A |
| | | $m$ | Mass | V, F | V, F, P | $\mathcal{U}(1,5)$ | $\mathcal{U}_{\log}(10^{-2}, 10^0)$ |
| | | $g$ | Gravitational acceleration | V, F | C, F, P | $\mathcal{U}(1,5)$ | $9.807 \times 10^0$ |
| | | $z$ | Height | V, F | V, F | $\mathcal{U}(1,5)$ | $\mathcal{U}_{\log}(10^{-2}, 10^0)$ |
| I.14.4 | $U = \frac{k_{\mathrm{spring}} x^2}{2}$ | $U$ | Elastic energy | V, F | V, F, P | N/A | N/A |
| | | $k_{\mathrm{spring}}$ | Spring constant | V, F | V, F, P | $\mathcal{U}(1,5)$ | $\mathcal{U}_{\log}(10^2, 10^4)$ |
| | | $x$ | Position | V, F | V, F | $\mathcal{U}(1,5)$ | $\mathcal{U}_{\log}(10^{-2}, 10^0)$ |
| I.18.12 | $\tau = rF\sin\theta$ | $\tau$ | Torque | V, F | V, F | N/A | N/A |
| | | $r$ | Distance | V, F | V, F, P | $\mathcal{U}(1,5)$ | $\mathcal{U}_{\log}(10^{-1}, 10^1)$ |
| | | $F$ | Force | V, F | V, F | $\mathcal{U}(1,5)$ | $\mathcal{U}_{\log}(10^{-1}, 10^1)$ |
| | | $\theta$ | Angle | V, F | V, F, NN | $\mathcal{U}(0,5)$ | $\mathcal{U}(0, 2\pi)$ |
| I.18.16 | $L = mrv\sin\theta$ | $L$ | Angular momentum | V, F | V, F | N/A | N/A |
| | | $m$ | Mass | V, F | V, F, P | $\mathcal{U}(1,5)$ | $\mathcal{U}_{\log}(10^{-1}, 10^1)$ |
| | | $r$ | Distance | V, F | V, F, P | $\mathcal{U}(1,5)$ | $\mathcal{U}_{\log}(10^{-1}, 10^1)$ |
| | | $v$ | Velocity | V, F | V, F, P | $\mathcal{U}(1,5)$ | $\mathcal{U}_{\log}(10^{-1}, 10^1)$ |
| | | $\theta$ | Angle | V, F | V, F, NN | $\mathcal{U}(1,5)$ | $\mathcal{U}(0, 2\pi)$ |
| I.25.13 | $V = \frac{q}{C}$ | $V$ | Voltage | V, F | V, F | N/A | N/A |
| | | $q$ | Electric charge | V, F | V, F | $\mathcal{U}(1,5)$ | $\mathcal{U}_{\log}(10^{-5}, 10^{-3})$ |
| | | $C$ | Electrostatic Capacitance | V, F | V, F, P | $\mathcal{U}(1,5)$ | $\mathcal{U}_{\log}(10^{-5}, 10^{-3})$ |
| I.26.2 | $n = \frac{\sin\theta_1}{\sin\theta_2}$ | $n$ | Relative refractive index | V, F | V, F, P | $\mathcal{U}(0,1)$ | N/A |
| | | $\theta_1$ | Refraction angle 1 | V, F | V, F | N/A | $\mathcal{U}(0, \frac{\pi}{2})$ |
| | | $\theta_2$ | Refraction angle 2 | V, F | V, F | $\mathcal{U}(1,5)$ | $\mathcal{U}(0, \frac{\pi}{2})$ |
| I.27.6 | $f = \frac{1}{\frac{1}{d_1} + \frac{n}{d_2}}$ | $f$ | Focal length | V, F | V, F | N/A | N/A |
| | | $d_1$ | Distance | V, F | V, F, P | $\mathcal{U}(1,5)$ | $\mathcal{U}_{\log}(10^{-3}, 10^{-1})$ |
| | | $n$ | Refractive index | V, F | V, F, P, | $\mathcal{U}(1,5)$ | $\mathcal{U}_{\log}(10^{-1}, 10^1)$ |
| | | $d_2$ | Distance | V, F | V, F, P | $\mathcal{U}(1,5)$ | $\mathcal{U}_{\log}(10^{-3}, 10^{-1})$ |

considers magnitude of sampling distributions (*Distributions* column in Tables 1 and S1 – S11) and increases the complexity when sampling values from wide range of distributions. We define the domain range as follows:

$$f_{\mathrm{range}}(\mathcal{S}) = \left| \log_{10} \left| \max_{s \in \mathcal{S}} s - \min_{s \in \mathcal{S}} s \right| \right|, \tag{1}$$

where $\mathcal{S}$ indicates a set of sampling domains (*distributions*) for a given symbolic regression problem.

As we will show in Section 5.3, these clusters represent problem difficulties at high level. For instance, these subsets will help the research community to shortly tune and/or perform sanity-check new approaches on the *Easy* set (30 problems) instead of using the whole datasets (120 problems). Figure 1 shows the three different distribution maps of our proposed datasets. *Easy*, *Medium*, and *Hard* sets consist of 30, 40, and 50 individual symbolic regression problems, respectively.

# 4 BENCHMARK

Besides the conventional metrics, we propose a new metric to discuss the performance of symbolic regression for scientific discovery in Section 4.1. Following the set of metrics, we design an evaluation framework of symbolic regression for scientific discovery.

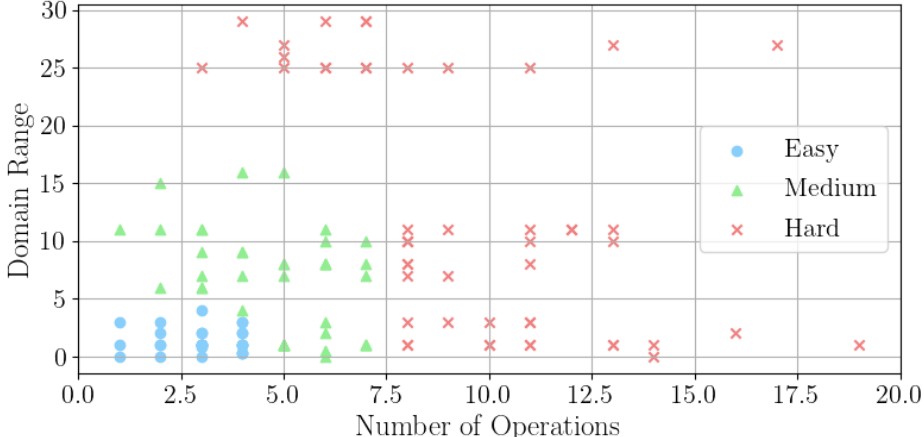

Figure 1: Distribution map of our proposed datasets based on three different subsets with respect to our complexity metrics. Data points at top right/bottom left indicate more/less complex problems.

$$f_{\text{true}} = \frac{4\pi\mu B}{h}$$

1. Substitute values

$\pi = 3.14159265358979$
$h = 6.626 \times 10^{-34}$

2. Convert to an equation tree

$= (8.32647716907439 \times 10^{-33})\mu B$

Mul

$C_1$   $X_1$   $X_2$

Figure 2: Example of preprocessing a true equation (III.7.38 in Table S1) in evaluation session. When converting to an equation tree, we replace constant values and variables with specific symbols *e.g.*, $8.32647716907439 \times 10^{-33} \rightarrow C_1, \mu \rightarrow X_1, B \rightarrow X_2$.

## 4.1 METRICS

In general, it would be difficult to define "accuracy" of symbolic regression models since we will compare its estimated equation to the ground truth equation and need criteria to determine whether or not it is "correct". La Cava et al. (2021) suggested a reasonable definition of symbolic solution, which is designed to capture symbolic regression models that differ from the true model by a constant or scalar. They also used $R^2$ score (Eq. 2) and define as accuracy the percentage of symbolic regression problems that a model meets $R^2 > \tau$, where $\tau$ is a threshold *e.g.*, $\tau = 0.999$ in (La Cava et al., 2021).

$$R^2 = \frac{\sum_j^N \left(f_{\text{pred}}\left(X_j\right) - f_{\text{true}}\left(X_j\right)\right)^2}{\sum_k^N \left(f_{\text{true}}\left(X_k\right) - \bar{y}\right)^2}, \tag{2}$$

where $N$ indicates the number of test samples (*i.e.*, the number of rows in the test dataset), and $\bar{y}$ is a mean of target outputs produced by $f_{\text{true}}$. $f_{\text{pred}}$ and $f_{\text{true}}$ are a trained SR model and a true model, respectively. However, these two metrics are still binary (correct or not) or require a threshold and does not explain how *structurally close* to the true equation the estimated one is. While a key feature of symbolic regression is its interpretability, there are no single evaluation metrics to take into account both the interpretability and how close to the true expression the estimated expression is.

To offer more flexibility and assess estimated equations in such a way, we propose the use of edit distance between estimated and ground truth equations, processing equations as trees. Although edit distance has been employed in different domains such as machine translation (Przybocki et al., 2006) (text-based edit distance), its primary use has been to study the search process for genetic programming approaches (O'Reilly, 1997; Burke et al., 2002; Nakai et al., 2013). Different from prior work, we propose a use of tree-based edit distance as a new metric of solution quality for SRSD.

For a pair of two trees, edit distance computes the minimum cost to transform one to another with a sequence of operations, each of which either 1) *inserts*, 2) *deletes*, or 3) *renames* a node. In this study, a node can be either a mathematical operation (*e.g.*, *add*, *exp* as symbols), a variable symbol, or a constant symbol. For the detail of the algorithm, we refer readers to (Zhang & Shasha, 1989).

As illustrated in Fig. 2, we preprocess equations by 1) substituting constant values *e.g.*, $\pi$ and Planck constant to the expression, and 2) converting the resulting expression to an equation tree that represents the preorder traversal of the equation with simplified symbols. It should be worth noting that before generating the equation tree, we simplify and convert equations to floating-point approximations by sympy Meurer et al. (2017), a Python library for symbolic mathematics. It helps us consistently map a given equation to the unique equation tree and compute edit distance between the true and estimated equation trees since our evaluation interest is in simplified expressions of the estimated equations rather than how SR models produced the equations. For instance, "$x + x + x$", "$4 * x - x$", and "$x + 2 * x$" will be simplified by sympy to "$3 * x$" and considered identical.

For edit distance, we use a method proposed by Zhang & Shasha (1989). Given that the range of edit distance values depends on complexity of equations, we normalize the distance in range of 0 to 1 as

$$\bar{d}(f_{\text{pred}}, f_{\text{true}}) = \min\left(1, \frac{d(f_{\text{pred}}, f_{\text{true}})}{|f_{\text{true}}|}\right), \qquad (3)$$

where $f_{\text{pred}}$ and $f_{\text{true}}$ are estimated and true equation trees, respectively. $d(f_{\text{pred}}, f_{\text{true}})$ is an edit distance between $f_{\text{pred}}$ and $f_{\text{true}}$. $|f_{\text{true}}|$ indicates the number of the tree nodes that compose an equation $f_{\text{true}}$. We note that this metric is designed to capture similarity between estimated and true equations, thus coefficient values themselves (*e.g.*, value of $C_1$ in Fig. 2) should not be important.

### 4.2 EVALUATION FRAMEWORK

For real datasets (assuming observed datasets), only tabular data are available for training and validation. (In practice, a test dataset does not include the true equation.) For benchmark purposes, true equations are provided as test data besides test tabular data.

For each problem, we use the validation tabular dataset and choose the best trained SR model $f_{\text{pred}}^*$ from $\mathcal{F}$, a set of the trained models by a given method respect to Eq. (4)

$$f_{\text{pred}}^* = \underset{f_{\text{pred}} \in \mathcal{F}}{\arg\min} \frac{1}{n} \sum_{i=1}^{n} \left| \frac{f_{\text{pred}}(X_i) - f_{\text{true}}(X_i)}{f_{\text{true}}(X_i)} \right|^2, \qquad (4)$$

where $X_i$ indicates the $i$-th row of the validation tabular dataset $X$.

Notice that while we proposed in Section 4.1 a normalized edit distance between estimated and true equation trees, such true equations will not be available in practice, especially when using symbolic regression methods for scientific discovery. For this reason, we use the geometrical distance between predicted values against a validation tabular dataset to choose the best model obtained through hyperparameter tuning. Using the best model per method, we compute the normalized edit distance to assess the method.

## 5 EXPERIMENTS

### 5.1 BASELINE METHODS

For baselines, we use the five best symbolic regression methods in SRBench (La Cava et al., 2021). Specifically, we choose gplearn (Koza & Poli, 2005), AFP (Schmidt & Lipson, 2011), AFP-FE (Schmidt & Lipson, 2009), AI Feynman (Udrescu et al., 2020), and DSR (Petersen et al., 2020), referring to the rankings of solution rate for the FSRD datasets in their study. We note that La Cava et al. (2021) also benchmark symbolic regression methods for black-box problems, whose true symbolic expressions are unknown, and other symbolic regression methods *e.g.*, Operon (Kommenda et al., 2020), SBP-GP (Virgolin et al., 2019), FEAT (La Cava et al., 2018), EPLEX (La Cava et al., 2019), and GP-GOMEA (Virgolin et al., 2021) outperform the five baseline methods we choose from their study, in terms of $R^2$-driven accuracy. However, we find solution rate more aligned with edit distance, thus we choose the five best symbolic regression methods in terms of solution rate

empirically shown for the FSRD datasets in SRBench (La Cava et al., 2021). In addition to the five existing symbolic regression baselines, we introduce Symbolic Transformer, a new baseline model with Transformer (Vaswani et al., 2017):

1. **gplearn** (Koza & Poli, 2005): a genetic programming based symbolic regression method published as a Python package `gplearn`.
2. **AFP** (Schmidt & Lipson, 2011): Age-fitness pareto optimization.
3. **AFP-FE** (Schmidt & Lipson, 2009): AFP optimization with fitness estimates.
4. **AI Feynman** (Udrescu et al., 2020): an iterative approach to generate symbolic regression to seek to fit data to formulas that are Pareto-optimal.
5. **DSR** (Petersen et al., 2020): reinforcement learning based deep symbolic regression.
6. **Symbolic Transformer (ST)**: our Transformer-based symbolic regression baseline.

For the details of the baseline models, we refer readers to the corresponding papers (Koza & Poli, 2005; Schmidt & Lipson, 2011; 2009; Udrescu et al., 2020; Petersen et al., 2020). We provide the details of Symbolic Transformer in Section C, including our pretraining strategy. While Symbolic Transformer itself is a new model, we note that the main contribution of this work lies in the datasets and benchmark of symbolic regression for scientific discovery. Our Transformer-based baseline method is simply inspired by recent advances in deep learning, specifically transformer-based high-performance, modern, and flexible models such as (Vaswani et al., 2017; Devlin et al., 2019; Dosovitskiy et al., 2020). Thus, the new model is not necessarily designed to show improvements over the existing Transformer-based symbolic regression models (Valipour et al., 2021; Biggio et al., 2021) including contemporary work such as (Kamienny et al., 2022).

## 5.2 RUNTIME CONSTRAINTS

The implementations of the baseline methods in Section 5.1 except Symbolic Transformer [5] do not use any GPUs. We run 600 high performance computing (HPC) jobs in total, using computing nodes in an HPC infrastructure, which have 5 - 20 assigned physical CPU cores, 30 - 120 GB RAM, and 720 GB local storage. Due to the properties of our HPC resource, we have some runtime constraints:

1. Since each HPC job is designed to run for up to 24 hours due to the limited resource, we run a job with a pair of a target tabular dataset and a symbolic regression method.
2. Given a pair of a dataset and a method, each of our HPC jobs runs up to 100 separate training sessions with different hyperparameter values.

## 5.3 RESULTS

In this section, we discuss the experimental results of our baseline methods, using the proposed SRSD datasets. Tables 2 and 3 show the performance of the symbolic regression baseline methods in terms of $R^2$-driven accuracy ($R^2 > 0.999$) and solution rate respectively, and both the metrics are used in SRBench (La Cava et al., 2021). According to the metrics, DSR significantly outperforms all the other baselines we considered. The DSR results also indicate difficulty levels of the three categories of our SRSD datasets, which looks aligned with our complexity-aware dataset categorization (Section 3.2.2).

Now we discuss the results using the normalized edit distance. Table 4 shows the results of the baseline methods in terms of the normalized edit distance. Interestingly, while the Symbolic Transformer performed the worst in Table 2 in terms of $R^2$-based accuracy, it achieved the best normalized edit distance for all the SRSD Easy, Medium, and Hard sets and significantly improved DSR. This trend also implies that the $R^2$-based accuracy does not always indicate how well the SR model can produce an equation that is *structurally close* to the true equation. We also confirmed that the difficulty level of each set of SRSD datasets is reflected to the overall trend in Tables 2 - 4.

We also performed a user study and how aligned with human judges the existing SR and new SRSD evaluation metrics are. The results show that the SRSD evaluation metric (NED) is more aligned with human judges than $R^2$ score. We refer readers to Section G for more details.

---

[5]We used an NVIDIA GeForce RTX 3070 Ti for pretraining Symbolic Transformer.

Table 2: Baseline results: accuracy ($R^2 > 0.999$) defined by La Cava et al. (2021).

| SRSD Datasets \ Method | gplearn | AFP | AFP-FE | AI Feynman | DSR | ST |
|---|---|---|---|---|---|---|
| Easy set (30 problems) | 6.67% | 20.0% | 23.3% | 33.3% | **60.0%** | 0.00% |
| Medium set (40 problems) | 7.50% | 2.50% | 2.50% | 5.00% | **42.5%** | 0.00% |
| Hard set (50 problems) | 2.00% | 4.00% | 4.00% | 8.00% | **30.0%** | 0.00% |

Table 3: Baseline results: solution rate defined by La Cava et al. (2021).

| SRSD Datasets \ Method | gplearn | AFP | AFP-FE | AI Feynman | DSR | ST |
|---|---|---|---|---|---|---|
| Easy set (30 problems) | 6.67% | 20.0% | 20.0% | 30.0% | **43.3%** | 16.7% |
| Medium set (40 problems) | 2.50% | 2.50% | 2.50% | 2.50% | **10.0%** | 5.00% |
| Hard set (50 problems) | 0.00% | 0.00% | 0.00% | **2.00%** | **2.00%** | 0.00% |

Table 4: Baseline results: our proposed normalized edit distances (the smaller the better).

| SRSD Datasets \ Method | gplearn | AFP | AFP-FE | AI Feynman | DSR | ST |
|---|---|---|---|---|---|---|
| Easy set (30 problems) | 0.876 | 0.703 | 0.712 | 0.646 | 0.551 | **0.435** |
| Medium set (40 problems) | 0.939 | 0.873 | 0.897 | 0.936 | 0.789 | **0.556** |
| Hard set (50 problems) | 0.978 | 0.960 | 0.956 | 0.930 | 0.833 | **0.704** |

## 6 LIMITATIONS

### 6.1 IMPLICIT FUNCTIONS

Symbolic regression generally has a limitation in inferring implicit functions, as the model infers a trivial constant function if there are no restrictions on variables. For example, $f(x, y) = 0$ is inferred as $0 = 0 \, \forall x, y$. This problem can be solved by applying the constraint that an inferred function should depend on at least two variables *e.g.*, inferring $f(x, y) = 0$ with $\frac{\partial f}{\partial x} \neq 0$ and $\frac{\partial f}{\partial y} \neq 0$, or by converting the function to an explicit form *e.g.*, $y = g(x)$. We converted some functions in the datasets into explicit forms and avoided the inverse trigonometric functions as described in Section 3.1.

### 6.2 DUMMY VARIABLES AND NOISE INJECTION

When applying machine learning to real-world problems, it is often true that 1) not all the observed features (variables in symbolic regression) are necessary to solve the problems, and 2) the observed values contain some noise. While we follow La Cava et al. (2021) and show experimental results for our SRSD datasets with noise-injected target variables in Section E, these aspects are not thoroughly discussed in this study, such discussions can be a separate paper built on this work and further engage studies of symbolic regression for scientific discovery.

## 7 CONCLUSION

In this work, we pointed out issues of existing datasets and benchmarks of symbolic regression for scientific discovery (SRSD). To address the issues, we proposed 1) 120 new SRSD datasets based on a set of physics formulas in FSRD (Udrescu & Tegmark, 2020) and 2) a new evaluation metric for SRSD to discuss the structural similarity between the true and estimated symbolic expressions (equations). We note that this study argues that the normalized edit distance is a metric not to take the place of existing SR metrics but to incorporate such metrics (*e.g.*, Tables 2 and 4). Besides the main contribution above, we proposed a Transformer-based symbolic regression baseline, named Symbolic Transformer that achieved the best normalized edit distance for the proposed SRSD datasets. To encourage the studies of SRSD, we provide our datasets and code with MIT License.

## (OPTIONAL) ETHICS STATEMENT

In this study, we recruited 23 human subjects and performed a user study (see Section G) with approval from an ethics review board. We performed the user study to discuss how meaningful our new SRSD evaluation metric is, compared to existing SR metrics. Subjects were asked to assess in scale of 1-to-5 how close to the true equation the estimated equation is, and in the user study we used only the human-based evaluations against equations. For this reason, we found no ethical concerns in the user study.

## (OPTIONAL) REPRODUCIBILITY STATEMENT

For reproducibility, we provide the following items in this paper and/or the supplementary material:

1. our datasets and code repository with MIT License and instructions to run our scripts, including all the five existing and one new baselines (See the supplementary material),

2. annotation policy and details of the 120 proposed datasets (See Section 3.2.1 and Tables 1 - S11),

3. architecture design and hyperparameters of Symbolic Transformer, a new SRSD baseline we introduced (See Section C and the supplementary material),

4. computing resources and runtime constraints in this study (See Section 5.2 and Footnote 5), and

5. hyperparameters used for the six baseline methods (See Sections B and C.2).

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

# A   OUR SRSD DATASETS: ADDITIONAL INFORMATION

This section provides additional information regarding our SRSD datasets. We created the datasets to discuss the performance of symbolic regression for scientific discovery (SRSD). Each of SRSD datasets consists of 10,000 samples and has train, val, and test splits with ratio of 8:1:1. We refer readers to Section 3 for details of the datasets. Tables S1 – S11 comprehensively summarize the differences between FSRD and our SRSD datasets. Note that the table of Easy set (part 1) is provided as Table 1 in Section 3.1. As described in Section 3.2.2, we categorized each of the 120 SRSD datasets into one of Easy, Medium, and Hard sets. The datasets and the documentations are provided as part of the supplementary material.

Table S1: Easy set of our proposed datasets (part 2). C: Constant, V: Variable, F: Float, I: Integer, P: Positive, N: Negative, NN: Non-Negative, I⋆: Integer treated as float due to the capacity of 32-bit integer, $\mathcal{U}$: Uniform distribution, $\mathcal{U}_{\log}$: Log-Uniform distribution.

| Eq. ID | Formula | Symbols | | Properties | | Distributions | |
| --- | --- | --- | --- | Original | Ours | Original | Ours |
| I.30.5 | $d = \frac{\lambda}{n \sin\theta}$ | $d$ | Interplanar distance | V, F | V, F, P | $\mathcal{U}(2,5)$ | N/A |
| | | $\lambda$ | Wavelength of X-ray | V, F | V, F, P | $\mathcal{U}(1,2)$ | $\mathcal{U}_{\log}(10^{-11}, 10^{-9})$ |
| | | $n$ | The number of phase difference | V, F | V, I, P | $\mathcal{U}(1,5)$ | $\mathcal{U}_{\log}(10^{0}, 10^{2})$ |
| | | $\theta$ | Incidence/Reflection angle | V, F | V, F | N/A | $\mathcal{U}(-2\pi, 2\pi)$ |
| I.43.16 | $v = \mu q \frac{V}{d}$ | $v$ | Velocity | V, F | V, F | N/A | N/A |
| | | $\mu$ | Ionic conductivity | V, F | V, F | $\mathcal{U}(1,5)$ | $\mathcal{U}_{\log}(10^{-6}, 10^{-4})$ |
| | | $q$ | Electric charge of ions | V, F | V, F | $\mathcal{U}(1,5)$ | $\mathcal{U}_{\log}(10^{-11}, 10^{-9})$ |
| | | $V$ | Voltage | V, F | V, F | $\mathcal{U}(1,5)$ | $\mathcal{U}_{\log}(10^{-1}, 10^{1})$ |
| | | $d$ | Distance | V, F | V, F, P | $\mathcal{U}(1,5)$ | $\mathcal{U}_{\log}(10^{-3}, 10^{-1})$ |
| I.47.23 | $c = \sqrt{\frac{\gamma P}{\rho}}$ | $c$ | Velocity of sound | V, F | V, F, P | N/A | N/A |
| | | $\gamma$ | Heat capacity ratio | V, F | V, F, P | $\mathcal{U}(1,5)$ | $\mathcal{U}(1,2)$ |
| | | $P$ | Atmospheric pressure | V, F | V, F, P | $\mathcal{U}(1,5)$ | $\mathcal{U}(0.5 \times 10^{-5}, 1.5 \times 10^{-5})$ |
| | | $\rho$ | Density of air | V, F | V, F, P | $\mathcal{U}(1,5)$ | $\mathcal{U}(1,2)$ |
| II.2.42 | $J = \kappa(T_2 - T_1)\frac{A}{d}$ | $J$ | Energy difference | V, F | V, F | N/A | N/A |
| | | $\kappa$ | Thermal conductivity | V, F | V, F, P | $\mathcal{U}(1,5)$ | $\mathcal{U}_{\log}(10^{-1}, 10^{1})$ |
| | | $T_2$ | Temperature | V, F | V, F, P | $\mathcal{U}(1,5)$ | $\mathcal{U}_{\log}(10^{1}, 10^{3})$ |
| | | $T_1$ | Temperature | V, F | V, F, P | $\mathcal{U}(1,5)$ | $\mathcal{U}_{\log}(10^{1}, 10^{3})$ |
| | | $A$ | Area | V, F | V, F, P | $\mathcal{U}(1,5)$ | $\mathcal{U}_{\log}(10^{-4}, 10^{-2})$ |
| | | $d$ | Length | V, F | V, F, P | $\mathcal{U}(1,5)$ | $\mathcal{U}_{\log}(10^{-2}, 10^{0})$ |
| II.3.24 | $h = \frac{W}{4\pi r^2}$ | $h$ | Heat flux | V, F | V, F | N/A | N/A |
| | | $W$ | Work | V, F | V, F | $\mathcal{U}(1,5)$ | $\mathcal{U}_{\log}(10^{0}, 10^{2})$ |
| | | $r$ | Distance | V, F | V, F, P | $\mathcal{U}(1,5)$ | $\mathcal{U}_{\log}(10^{-2}, 10^{0})$ |
| II.4.23 | $\phi = \frac{q}{4\pi\epsilon r}$ | $\phi$ | Electric potential | V, F | V, F | N/A | N/A |
| | | $q$ | Electric charge | V, F | V, F | $\mathcal{U}(1,5)$ | $\mathcal{U}_{\log}(10^{-3}, 10^{-1})$ |
| | | $\epsilon$ | Vacuum permittivity | V, F | C, F, P | $\mathcal{U}(1,5)$ | $8.854 \times 10^{-12}$ |
| | | $r$ | Distance | V, F | V, F, P | $\mathcal{U}(1,5)$ | $\mathcal{U}_{\log}(10^{-2}, 10^{0})$ |
| II.8.31 | $u = \frac{\epsilon E^2}{2}$ | $u$ | Energy | V, F | V, F | N/A | N/A |
| | | $\epsilon$ | Vacuum permittivity | V, F | C, F, P | $\mathcal{U}(1,5)$ | $8.854 \times 10^{-12}$ |
| | | $E$ | Magnitude of electric field | V, F | V, F, P | $\mathcal{U}(1,5)$ | $\mathcal{U}_{\log}(10^{1}, 10^{3})$ |
| II.10.9 | $E = \frac{\sigma_{\text{free}}}{\epsilon}\frac{1}{1+\chi}$ | $E$ | Electric field | V, F | V, F | N/A | N/A |
| | | $\sigma_{\text{free}}$ | Surface charge | V, F | V, F | $\mathcal{U}(1,5)$ | $\mathcal{U}_{\log}(10^{-3}, 10^{-1})$ |
| | | $\epsilon$ | Vacuum permittivity | V, F | C, F, P | $\mathcal{U}(1,5)$ | $8.854 \times 10^{-12}$ |
| | | $\chi$ | Electric susceptibility | V, F | V, F, P | $\mathcal{U}(1,5)$ | $\mathcal{U}_{\log}(10^{0}, 10^{2})$ |
| II.13.17 | $B = \frac{1}{4\pi\epsilon c^2}\frac{2I}{r}$ | $B$ | The magnitude of the magnetic field | V, F | V, F | N/A | N/A |
| | | $\epsilon$ | Vacuum permittivity | V, F | C, F, P | $\mathcal{U}(1,5)$ | $8.854 \times 10^{-12}$ |
| | | $c$ | Speed of light | V, F | C, F, P | $\mathcal{U}(1,5)$ | $2.998 \times 10^{8}$ |
| | | $I$ | Electric current | V, F | V, F | $\mathcal{U}(1,5)$ | $\mathcal{U}_{\log}(10^{-3}, 10^{-1})$ |
| | | $r$ | Radius | V, F | V, F, P | $\mathcal{U}(1,5)$ | $\mathcal{U}_{\log}(10^{-3}, 10^{-1})$ |
| II.15.4 | $U = -\mu B \cos\theta$ | $U$ | Energy from magnetic field | V, F | V, F | N/A | N/A |
| | | $\mu$ | Magnetic dipole moment | V, F | V, F | $\mathcal{U}(1,5)$ | $\mathcal{U}_{\log}(10^{-25}, 10^{-23})$ |
| | | $B$ | Magnetic field strength | V, F | V, F | $\mathcal{U}(1,5)$ | $\mathcal{U}_{\log}(10^{-3}, 10^{-1})$ |
| | | $\theta$ | Angle | V, F | V, F, NN | $\mathcal{U}(1,5)$ | $\mathcal{U}(0, 2\pi)$ |

Table S2: Easy set of our proposed datasets (part 3).

| Eq. ID | Formula | | Symbols | Properties | | Distributions | |
|--------|---------|--|---------|------------|--|---------------|--|
| | | | | Original | Ours | Original | Ours |
| II.15.5 | $U = -pE\cos\theta$ | $U$ | Energy | V, F | V, F | N/A | N/A |
| | | $p$ | Electric dipole moment | V, F | V, F | $\mathcal{U}(1,5)$ | $\mathcal{U}_{\log}(10^{-22}, 10^{-20})$ |
| | | $E$ | Magnitude of electric field | V, F | V, F | $\mathcal{U}(1,5)$ | $\mathcal{U}_{\log}(10^{1}, 10^{3})$ |
| | | $\theta$ | Angle | V, F | V, F | $\mathcal{U}(1,5)$ | $\mathcal{U}(0, 2\pi)$ |
| II.27.16 | $S = \epsilon c E^2$ | $S$ | Radiant intensity | V, F | V, F | N/A | N/A |
| | | $\epsilon$ | Vacuum permittivity | V, F | C, F, P | $\mathcal{U}(1,5)$ | $8.854 \times 10^{-12}$ |
| | | $c$ | Speed of light | V, F | C, F, P | $\mathcal{U}(1,5)$ | $2.998 \times 10^{8}$ |
| | | $E$ | Magnitude of electric field | V, F | V, F, P | $\mathcal{U}(1,5)$ | $\mathcal{U}_{\log}(10^{-1}, 10^{1})$ |
| II.27.18 | $u = \epsilon E^2$ | $u$ | Energy density | V, F | V, F, P | N/A | N/A |
| | | $\epsilon$ | Vacuum permittivity | V, F | C, F, P | $\mathcal{U}(1,5)$ | $8.854 \times 10^{-12}$ |
| | | $E$ | Magnitude of electric field | V, F | V, F, P | $\mathcal{U}(1,5)$ | $\mathcal{U}_{\log}(10^{-1}, 10^{1})$ |
| II.34.11 | $\omega = g\frac{qB}{2m}$ | $\omega$ | Angular frequency | V, F | V, F | N/A | N/A |
| | | $g$ | g-factor | V, F | V, F | $\mathcal{U}(1,5)$ | $\mathcal{U}(-1, 1)$ |
| | | $q$ | Electric charge | V, F | V, F | $\mathcal{U}(1,5)$ | $\mathcal{U}_{\log}(10^{-11}, 10^{-9})$ |
| | | $B$ | Magnetic field strength | V, F | V, F | $\mathcal{U}(1,5)$ | $\mathcal{U}_{\log}(10^{-9}, 10^{-7})$ |
| | | $m$ | Mass | V, F | V, F, P | $\mathcal{U}(1,5)$ | $\mathcal{U}_{\log}(10^{-30}, 10^{-28})$ |
| II.34.29b | $U = 2\pi g\mu B\frac{J_z}{h}$ | $U$ | Energy | V, F | V, F, P | N/A | N/A |
| | | $g$ | g-factor | V, F | V, F | $\mathcal{U}(1,5)$ | $\mathcal{U}(-1, 1)$ |
| | | $\mu$ | Bohr magneton | V, F | C, F, P | $\mathcal{U}(1,5)$ | $9.2740100783 \times 10^{-24}$ |
| | | $B$ | Magnetic field strength | V, F | V, F | $\mathcal{U}(1,5)$ | $\mathcal{U}_{\log}(10^{-3}, 10^{-1})$ |
| | | $J_z$ | Element of angular momentum | V, F | V, F | $\mathcal{U}(1,5)$ | $\mathcal{U}_{\log}(10^{-26}, 10^{-22})$ |
| | | $h$ | Planck constant | V, F | C, F, P | $\mathcal{U}(1,5)$ | $6.626 \times 10^{-34}$ |
| II.38.3 | $F = YA\frac{\Delta l}{l}$ | $F$ | Force | V, F | V, F | N/A | N/A |
| | | $Y$ | Young's modulus | V, F | V, F, P | $\mathcal{U}(1,5)$ | $\mathcal{U}_{\log}(10^{-1}, 10^{1})$ |
| | | $A$ | Area | V, F | V, F, P | $\mathcal{U}(1,5)$ | $\mathcal{U}_{\log}(10^{-4}, 10^{-2})$ |
| | | $\delta l$ | Displacement | V, F | V, F | $\mathcal{U}(1,5)$ | $\mathcal{U}_{\log}(10^{-3}, 10^{-1})$ |
| | | $l$ | Length | V, F | V, F, P | $\mathcal{U}(1,5)$ | $\mathcal{U}_{\log}(10^{-2}, 10^{0})$ |
| II.38.14 | $\mu = \frac{Y}{2(1+\sigma)}$ | $\mu$ | Rigidity modulus | V, F | V, F, P | N/A | N/A |
| | | $Y$ | Young's modulus | V, F | V, F, P | $\mathcal{U}(1,5)$ | $\mathcal{U}_{\log}(10^{-1}, 10^{1})$ |
| | | $\sigma$ | Poisson coefficient | V, F | V, F, P | $\mathcal{U}(1,5)$ | $\mathcal{U}_{\log}(10^{-2}, 10^{0})$ |
| III.7.38 | $\omega = \frac{4\pi\mu B}{h}$ | $\omega$ | Precession frequency | V, F | V, F | N/A | N/A |
| | | $\mu$ | Magnetic moment | V, F | V, F | $\mathcal{U}(1,5)$ | $\mathcal{U}_{\log}(10^{-11}, 10^{-9})$ |
| | | $B$ | Magnetic flux density | V, F | V, F | $\mathcal{U}(1,5)$ | $\mathcal{U}_{\log}(10^{-3}, 10^{-1})$ |
| | | $h$ | Planck constant | V, F | C, F, P | $\mathcal{U}(1,5)$ | $6.626 \times 10^{-34}$ |
| III.12.43 | $J = \frac{mh}{2\pi}$ | $J$ | Variable | V, F | V, F | N/A | N/A |
| | | $m$ | Spin state | V, F | V, I,NN | $\mathcal{U}(1,5)$ | $\mathcal{U}_{\log}(10^{0}, 10^{2})$ |
| | | $h$ | Planck constant | V, F | C, F, P | $\mathcal{U}(1,5)$ | $6.626 \times 10^{-34}$ |
| III.15.27 | $k = \frac{2\pi}{Nb}s$ | $k$ | Wavenumber | V, F | V, F | N/A | N/A |
| | | $s$ | Parameter of state | V, F | V, I | $\mathcal{U}(1,5)$ | $\mathcal{U}_{\log}(10^{0}, 10^{2})$ |
| | | $N$ | Number of atoms | V, F | V, I,P | $\mathcal{U}(1,5)$ | $\mathcal{U}_{\log}(10^{0}, 10^{2})$ |
| | | $b$ | Lattice constant | V, F | V, F, P | $\mathcal{U}(1,5)$ | $\mathcal{U}_{\log}(10^{-10}, 10^{-8})$ |

Table S3: Medium set of our proposed datasets (part 1).

| Eq. ID | Formula | Symbols | | Properties | | Distributions | |
|---|---|---|---|---|---|---|---|
| | | | | Original | Ours | Original | Ours |
| I.8.14 | $d = \sqrt{(x_2 - x_1)^2 + (y_2 - y_1)^2}$ | $d$ | Distance | V, F | V, F, NN | N/A | N/A |
| | | $x_2$ | Position | V, F | V, F | $\mathcal{U}(1,5)$ | $\mathcal{U}_{\log}(10^{-1}, 10^1)$ |
| | | $x_1$ | Position | V, F | V, F | $\mathcal{U}(1,5)$ | $\mathcal{U}_{\log}(10^{-1}, 10^1)$ |
| | | $y_2$ | Position | V, F | V, F | $\mathcal{U}(1,5)$ | $\mathcal{U}_{\log}(10^{-1}, 10^1)$ |
| | | $y_1$ | Position | V, F | V, F | $\mathcal{U}(1,5)$ | $\mathcal{U}_{\log}(10^{-1}, 10^1)$ |
| I.10.7 | $m = \dfrac{m_0}{\sqrt{1 - \frac{v^2}{c^2}}}$ | $m$ | Mass | V, F | V, F, P | N/A | N/A |
| | | $m_0$ | Invariant mass | V, F | V, F, P | $\mathcal{U}(1,5)$ | $\mathcal{U}_{\log}(10^{-1}, 10^1)$ |
| | | $v$ | Velocity | V, F | V, F, P | $\mathcal{U}(1,2)$ | $\mathcal{U}_{\log}(10^5, 10^8)$ |
| | | $c$ | Speed of light | V, F | C, F, P | $\mathcal{U}(3,10)$ | $2.998 \times 10^8$ |
| I.11.19 | $A = x_1 y_1 + x_2 y_2 + x_3 y_3$ | $A$ | Inner product | V, F | V, F | N/A | N/A |
| | | $x_1$ | Element of a vector | V, F | V, F | $\mathcal{U}(1,5)$ | $\mathcal{U}_{\log}(10^{-1}, 10^1)$ |
| | | $y_1$ | Element of a vector | V, F | V, F | $\mathcal{U}(1,5)$ | $\mathcal{U}_{\log}(10^{-1}, 10^1)$ |
| | | $x_2$ | Element of a vector | V, F | V, F | $\mathcal{U}(1,5)$ | $\mathcal{U}_{\log}(10^{-1}, 10^1)$ |
| | | $y_2$ | Element of a vector | V, F | V, F | $\mathcal{U}(1,5)$ | $\mathcal{U}_{\log}(10^{-1}, 10^1)$ |
| | | $x_3$ | Element of a vector | V, F | V, F | $\mathcal{U}(1,5)$ | $\mathcal{U}_{\log}(10^{-1}, 10^1)$ |
| | | $y_3$ | Element of a vector | V, F | V, F | $\mathcal{U}(1,5)$ | $\mathcal{U}_{\log}(10^{-1}, 10^1)$ |
| I.12.2 | $F = \dfrac{q_1 q_2}{4\pi\epsilon r^2}$ | $F$ | Electrostatic force | V, F | V, F | N/A | N/A |
| | | $q_1$ | Electric charge | V, F | V, F | $\mathcal{U}(1,5)$ | $\mathcal{U}_{\log}(10^{-3}, 10^{-1})$ |
| | | $q_2$ | Electric charge | V, F | V, F | $\mathcal{U}(1,5)$ | $\mathcal{U}_{\log}(10^{-3}, 10^{-1})$ |
| | | $r$ | Distance | V, F | V, F, P | $\mathcal{U}(1,5)$ | $\mathcal{U}_{\log}(10^{-2}, 10^0)$ |
| | | $\epsilon$ | Vacuum permittivity | V, F | C, F, P | $\mathcal{U}(1,5)$ | $8.854 \times 10^{-12}$ |
| I.12.11 | $F = q\left(E + Bv\sin(\theta)\right)$ | $F$ | Force | V, F | V, F | N/A | N/A |
| | | $q$ | Electric charge | V, F | V, F | $\mathcal{U}(1,5)$ | $\mathcal{U}_{\log}(10^{-1}, 10^1)$ |
| | | $E$ | Electric field | V, F | V, F | $\mathcal{U}(1,5)$ | $\mathcal{U}_{\log}(10^{-1}, 10^1)$ |
| | | $B$ | Magnetic field strength | V, F | V, F, P | $\mathcal{U}(1,5)$ | $\mathcal{U}_{\log}(10^{-1}, 10^1)$ |
| | | $v$ | Velocity | V, F | V, F, P | $\mathcal{U}(1,5)$ | $\mathcal{U}_{\log}(10^{-1}, 10^1)$ |
| | | $\theta$ | Angle | V, F | V, F, NN | $\mathcal{U}(1,5)$ | $\mathcal{U}(0, 2\pi)$ |
| I.13.4 | $K = \frac{1}{2}m(v^2 + u^2 + w^2)$ | $K$ | Kinetic energy | V, F | V, F, P | N/A | N/A |
| | | $m$ | Mass | V, F | V, F, P | $\mathcal{U}(1,5)$ | $\mathcal{U}_{\log}(10^{-2}, 10^0)$ |
| | | $v$ | Element of velocity | V, F | V, F, P | $\mathcal{U}(1,5)$ | $\mathcal{U}_{\log}(10^{-1}, 10^1)$ |
| | | $u$ | Element of velocity | V, F | V, F, P | $\mathcal{U}(1,5)$ | $\mathcal{U}_{\log}(10^{-1}, 10^1)$ |
| | | $w$ | Element of velocity | V, F | V, F, P | $\mathcal{U}(1,5)$ | $\mathcal{U}_{\log}(10^{-1}, 10^1)$ |
| I.13.12 | $U = Gm_1 m_2\left(\frac{1}{r_2} - \frac{1}{r_1}\right)$ | $U$ | Potential energy | V, F | V, F, P | N/A | N/A |
| | | $G$ | Gravitational constant | V, F | C, F, P | $\mathcal{U}(1,5)$ | $6.674 \times 10^{-11}$ |
| | | $m_1$ | Mass (The Earth) | V, F | V, F, P | $\mathcal{U}(1,5)$ | $\mathcal{U}_{\log}(10^{-2}, 10^0)$ |
| | | $m_2$ | Mass | V, F | V, F, P | $\mathcal{U}(1,5)$ | $\mathcal{U}_{\log}(10^{-2}, 10^0)$ |
| | | $r_2$ | Distance | V, F | V, F, P | $\mathcal{U}(1,5)$ | $\mathcal{U}_{\log}(10^{-2}, 10^0)$ |
| | | $r_1$ | Distance | V, F | V, F, P | $\mathcal{U}(1,5)$ | $\mathcal{U}_{\log}(10^{-2}, 10^0)$ |
| I.15.10 | $p = \dfrac{m_0 v}{\sqrt{1 - v^2/c^2}}$ | $p$ | Relativistic mass | V, F | V, F, P | N/A | N/A |
| | | $m_0$ | Rest Mass | V, F | V, F, P | $\mathcal{U}(1,5)$ | $\mathcal{U}_{\log}(10^{-2}, 10^0)$ |
| | | $v$ | Velocity | V, F | V, F | $\mathcal{U}(1,2)$ | $\mathcal{U}_{\log}(10^5, 10^7)$ |
| | | $c$ | Speed of light | V, F | C, F, P | $\mathcal{U}(3,10)$ | $2.998 \times 10^8$ |
| I.16.6 | $v_1 = \dfrac{u+v}{1+uv/c^2}$ | $v_1$ | Velocity | V, F | V, F | N/A | N/A |
| | | $u$ | Velocity | V, F | V, F | $\mathcal{U}(1,5)$ | $\mathcal{U}_{\log}(10^6, 10^8)$ |
| | | $v$ | Velocity | V, F | V, F | $\mathcal{U}(1,5)$ | $\mathcal{U}_{\log}(10^6, 10^8)$ |
| | | $c$ | Speed of light | V, F | C, F, P | $\mathcal{U}(1,5)$ | $2.998 \times 10^8$ |
| I.18.4 | $r = \dfrac{m_1 r_1 + m_2 r_2}{m_1 + m_2}$ | $r$ | Center of gravity | V, F | V, F | N/A | N/A |
| | | $m_1$ | Mass | V, F | V, F, P | $\mathcal{U}(1,5)$ | $\mathcal{U}_{\log}(10^{-1}, 10^1)$ |
| | | $r_1$ | Position | V, F | V, F | $\mathcal{U}(1,5)$ | $\mathcal{U}_{\log}(10^{-1}, 10^1)$ |
| | | $m_2$ | Mass | V, F | V, F, P | $\mathcal{U}(1,5)$ | $\mathcal{U}_{\log}(10^{-1}, 10^1)$ |
| | | $r_2$ | Position | V, F | V, F | $\mathcal{U}(1,5)$ | $\mathcal{U}_{\log}(10^{-1}, 10^1)$ |

Table S4: Medium set of our proposed datasets (part 2).

| Eq. ID | Formula | Symbols | | Properties | | Distributions | |
|---|---|---|---|---|---|---|---|
| | | | | Original | Ours | Original | Ours |
| I.24.6 | $E = \frac{1}{4}m(\omega^2 + \omega_0^2)x^2$ | $E$ | Energy | V, F | V, F, P | N/A | N/A |
| | | $m$ | Mass | V, F | V, F, P | $\mathcal{U}(1,3)$ | $\mathcal{U}_{\log}(10^{-1}, 10^1)$ |
| | | $\omega$ | Angular velocity | V, F | V, F | $\mathcal{U}(1,3)$ | $\mathcal{U}_{\log}(10^{-1}, 10^1)$ |
| | | $\omega_0$ | Angular velocity | V, F | V, F | $\mathcal{U}(1,3)$ | $\mathcal{U}_{\log}(10^{-1}, 10^1)$ |
| | | $x$ | Position | V, F | V, F | $\mathcal{U}(1,3)$ | $\mathcal{U}_{\log}(10^{-1}, 10^1)$ |
| I.29.4 | $k = \frac{\omega}{c}$ | $k$ | Wavenumber | V, F | V, F, P | N/A | N/A |
| | | $\omega$ | Frequency of electromagnetic waves | V, F | V, F, P | $\mathcal{U}(1,10)$ | $\mathcal{U}_{\log}(10^9, 10^{11})$ |
| | | $c$ | Speed of light | V, F | C, F, P | $\mathcal{U}(1,10)$ | $2.998 \times 10^8$ |
| I.32.5 | $P = \frac{q^2 a^2}{6\pi\epsilon c^3}$ | $P$ | Radiant energy | V, F | V, F, P | N/A | N/A |
| | | $q$ | Electric charge | V, F | V, F | $\mathcal{U}(1,5)$ | $\mathcal{U}_{\log}(10^{-3}, 10^{-1})$ |
| | | $a$ | Magnitude of direction vector | V, F | V, F, P | $\mathcal{U}(1,5)$ | $\mathcal{U}_{\log}(10^5, 10^7)$ |
| | | $\epsilon$ | Vacuum permittivity | V, F | C, F, P | $\mathcal{U}(1,5)$ | $8.854 \times 10^{-12}$ |
| | | $c$ | Speed of light | V, F | C, F, P | $\mathcal{U}(1,5)$ | $2.998 \times 10^8$ |
| I.34.8 | $\omega = \frac{qvB}{p}$ | $\omega$ | Angular velocity | V, F | V, F | N/A | N/A |
| | | $q$ | Electric charge | V, F | V, F | $\mathcal{U}(1,5)$ | $\mathcal{U}_{\log}(10^{-11}, 10^{-9})$ |
| | | $v$ | Velocity | V, F | V, F | $\mathcal{U}(1,5)$ | $\mathcal{U}_{\log}(10^5, 10^7)$ |
| | | $B$ | Magnetic field | V, F | V, F | $\mathcal{U}(1,5)$ | $\mathcal{U}_{\log}(10^1, 10^3)$ |
| | | $p$ | Angular momentum | V, F | V, F | $\mathcal{U}(1,5)$ | $\mathcal{U}_{\log}(10^9, 10^{11})$ |
| I.34.10 | $\omega = \frac{\omega_0}{1-v/c}$ | $\omega$ | Frequency of electromagnetic waves | V, F | V, F, P | N/A | N/A |
| | | $\omega_0$ | Frequency of electromagnetic waves | V, F | V, F, P | $\mathcal{U}(1,5)$ | $\mathcal{U}_{\log}(10^9, 10^{11})$ |
| | | $v$ | Velocity | V, F | V, F | $\mathcal{U}(1,2)$ | $\mathcal{U}_{\log}(10^5, 10^7)$ |
| | | $c$ | Speed of light | V, F | C, F, P | $\mathcal{U}(3,10)$ | $2.998 \times 10^8$ |
| I.34.27 | $W = \frac{h}{2\pi}\omega$ | $W$ | Energy | V, F | V, F, P | N/A | N/A |
| | | $h$ | Planck constant | V, F | C, F, P | $\mathcal{U}(1,5)$ | $6.626 \times 10^{-34}$ |
| | | $\omega$ | Frequency of electromagnetic waves | V, F | V, F, P | $\mathcal{U}(1,5)$ | $\mathcal{U}_{\log}(10^9, 10^{11})$ |
| I.38.12 | $r = 4\pi\epsilon\frac{(h/(2\pi))^2}{mq^2}$ | $r$ | Bohr radius | V, F | V, F, P | N/A | N/A |
| | | $\epsilon$ | Vacuum permittivity | V, F | C, F, P | $\mathcal{U}(1,5)$ | $8.854 \times 10^{-12}$ |
| | | $h$ | Planck constant | V, F | C, F, P | $\mathcal{U}(1,5)$ | $6.626 \times 10^{-34}$ |
| | | $m$ | Mass | V, F | V, F, P | $\mathcal{U}(1,5)$ | $\mathcal{U}_{\log}(10^{-28}, 10^{-26})$ |
| | | $q$ | Electric charge | V, F | V, F | $\mathcal{U}(1,5)$ | $\mathcal{U}_{\log}(10^{-11}, 10^{-9})$ |
| I.39.10 | $U = \frac{3}{2}PV$ | $U$ | Internal energy | V, F | V, F, P | N/A | N/A |
| | | $P$ | Pressure | V, F | V, F, P | $\mathcal{U}(1,5)$ | $\mathcal{U}_{\log}(10^4, 10^6)$ |
| | | $V$ | Volume | V, F | V, F, P | $\mathcal{U}(1,5)$ | $\mathcal{U}_{\log}(10^{-5}, 10^{-3})$ |
| I.39.11 | $U = \frac{PV}{\gamma-1}$ | $U$ | Energy | V, F | V, F | N/A | N/A |
| | | $\gamma$ | Heat capacity ratio | V, F | V, F, P | $\mathcal{U}(2,5)$ | $\mathcal{U}(1,2)$ |
| | | $P$ | Pressure | V, F | V, F, P | $\mathcal{U}(1,5)$ | $\mathcal{U}_{\log}(10^4, 10^6)$ |
| | | $V$ | Volume | V, F | V, F, P | $\mathcal{U}(1,5)$ | $\mathcal{U}_{\log}(10^{-5}, 10^{-3})$ |
| I.43.31 | $D = \mu kT$ | $D$ | Diffusion coefficient | V, F | V, F, P | N/A | N/A |
| | | $\mu$ | Viscosity | V, F | V, F, P | $\mathcal{U}(1,5)$ | $\mathcal{U}_{\log}(10^{13}, 10^{15})$ |
| | | $k$ | Boltzmann constant | V, F | C, F, P | $\mathcal{U}(1,5)$ | $1.381 \times 10^{-23}$ |
| | | $T$ | Temperature | V, F | V, F, P | $\mathcal{U}(1,5)$ | $\mathcal{U}_{\log}(10^1, 10^3)$ |

Table S5: Medium set of our proposed datasets (part 3).

| Eq. ID | Formula | Symbols | | Properties | | Distributions | |
|---|---|---|---|---|---|---|---|
| | | | | Original | Ours | Original | Ours |
| I.43.43 | $\kappa = \frac{1}{\gamma - 1} \frac{kv}{\sigma_c}$ | $\kappa$ | Thermal conductivity | V, F | V, F, P | N/A | N/A |
| | | $\gamma$ | Heat capacity ratio | V, F | V, F, P | $\mathcal{U}(2,5)$ | $\mathcal{U}(1,2)$ |
| | | $k$ | Boltzmann constant | V, F | C, F, P | $\mathcal{U}(1,5)$ | $1.381 \times 10^{-23}$ |
| | | $v$ | Velocity | V, F | V, F, P | $\mathcal{U}(1,5)$ | $\mathcal{U}_{\log}(10^2, 10^4)$ |
| | | $\sigma_c$ | Molecular collision cross section | V, F | V, F, P | $\mathcal{U}(1,5)$ | $\mathcal{U}_{\log}(10^{-21}, 10^{-19})$ |
| I.48.2 | $E = \frac{mc^2}{\sqrt{1 - v^2/c^2}}$ | $E$ | Energy | V, F | V, F, P | N/A | N/A |
| | | $m$ | Mass | V, F | V, F, P | $\mathcal{U}(1,5)$ | $\mathcal{U}_{\log}(10^{-29}, 10^{-27})$ |
| | | $c$ | Speed of light | V, F | C, F, P | $\mathcal{U}(3,10)$ | $2.998 \times 10^8$ |
| | | $v$ | Velocity | V, F | V, F, P | $\mathcal{U}(1,2)$ | $\mathcal{U}_{\log}(10^6, 10^8)$ |
| II.6.11 | $\phi = \frac{1}{4\pi\epsilon} \frac{p\cos\theta}{r^2}$ | $\phi$ | Electric potential | V, F | V, F | N/A | N/A |
| | | $\epsilon$ | Vacuum permittivity | V, F | C, F, P | $\mathcal{U}(1,3)$ | $8.854 \times 10^{-12}$ |
| | | $p$ | Electric dipole moment | V, F | V, F | $\mathcal{U}(1,3)$ | $\mathcal{U}_{\log}(10^{-22}, 10^{-20})$ |
| | | $\theta$ | Angle | V, F | V, F, NN | $\mathcal{U}(1,3)$ | $\mathcal{U}(0, 2\pi)$ |
| | | $r$ | Distance | V, F | V, F, P | $\mathcal{U}(1,3)$ | $\mathcal{U}_{\log}(10^{-10}, 10^{-8})$ |
| II.8.7 | $U = \frac{3}{5} \frac{Q^2}{4\pi\epsilon a}$ | $U$ | Energy | V, F | V, F | N/A | N/A |
| | | $Q$ | Electric charge | V, F | V, F | $\mathcal{U}(1,5)$ | $\mathcal{U}_{\log}(10^{-11}, 10^{-9})$ |
| | | $\epsilon$ | Vacuum permittivity | V, F | C, F, P | $\mathcal{U}(1,5)$ | $8.854 \times 10^{-12}$ |
| | | $a$ | Radius | V, F | V, F, P | $\mathcal{U}(1,5)$ | $\mathcal{U}_{\log}(10^{-12}, 10^{-10})$ |
| II.11.3 | $x = \frac{qE}{m(\omega_0^2 - \omega^2)}$ | $x$ | Position | V, F | V, F | N/A | N/A |
| | | $q$ | Electric charge | V, F | V, F | $\mathcal{U}(1,3)$ | $\mathcal{U}_{\log}(10^{-11}, 10^{-9})$ |
| | | $E$ | Magnitude of electric field | V, F | V, F, P | $\mathcal{U}(1,3)$ | $\mathcal{U}_{\log}(10^{-9}, 10^{-7})$ |
| | | $m$ | Mass | V, F | V, F, P | $\mathcal{U}(1,3)$ | $\mathcal{U}_{\log}(10^{-28}, 10^{-26})$ |
| | | $\omega_0$ | Angular velocity | V, F | V, F | $\mathcal{U}(3,5)$ | $\mathcal{U}_{\log}(10^9, 10^{11})$ |
| | | $\omega$ | Angular velocity | V, F | V, F | $\mathcal{U}(1,2)$ | $\mathcal{U}_{\log}(10^9, 10^{11})$ |
| II.21.32 | $\phi = \frac{q}{4\pi\epsilon r(1 - v/c)}$ | $\phi$ | Electric potential | V, F | V, F | N/A | N/A |
| | | $q$ | Electric charge | V, F | V, F | $\mathcal{U}(1,5)$ | $\mathcal{U}_{\log}(10^{-3}, 10^{-1})$ |
| | | $\epsilon$ | Vacuum permittivity | V, F | C, F, P | $\mathcal{U}(1,5)$ | $8.854 \times 10^{-12}$ |
| | | $r$ | Distance | V, F | V, F, P | $\mathcal{U}(1,5)$ | $\mathcal{U}_{\log}(10^0, 10^2)$ |
| | | $v$ | Velocity | V, F | V, F, P | $\mathcal{U}(1,2)$ | $\mathcal{U}_{\log}(10^5, 10^7)$ |
| | | $c$ | Speed of light | V, F | C, F, P | $\mathcal{U}(3,10)$ | $2.998 \times 10^8$ |
| II.34.2 | $\mu = \frac{qvr}{2}$ | $\mu$ | Magnetic moment | V, F | V, F | N/A | N/A |
| | | $q$ | Electric charge | V, F | V, F | $\mathcal{U}(1,5)$ | $\mathcal{U}_{\log}(10^{-11}, 10^{-9})$ |
| | | $v$ | Velocity | V, F | V, F | $\mathcal{U}(1,5)$ | $\mathcal{U}_{\log}(10^5, 10^7)$ |
| | | $r$ | Radius | V, F | V, F, P | $\mathcal{U}(1,5)$ | $\mathcal{U}_{\log}(10^{-11}, 10^{-9})$ |
| II.34.2a | $I = \frac{qv}{2\pi r}$ | $I$ | Electric Current | V, F | V, F | N/A | N/A |
| | | $q$ | Electric charge | V, F | V, F | $\mathcal{U}(1,5)$ | $\mathcal{U}_{\log}(10^{-11}, 10^{-9})$ |
| | | $v$ | Velocity | V, F | V, F | $\mathcal{U}(1,5)$ | $\mathcal{U}_{\log}(10^5, 10^7)$ |
| | | $r$ | Radius | V, F | V, F, P | $\mathcal{U}(1,5)$ | $\mathcal{U}_{\log}(10^{-11}, 10^{-9})$ |
| II.34.29a | $\mu = \frac{qh}{4\pi m}$ | $\mu$ | Bohr magneton | V, F | V, F | N/A | N/A |
| | | $q$ | Electric charge | V, F | V, F | $\mathcal{U}(1,5)$ | $\mathcal{U}_{\log}(10^{-11}, 10^{-9})$ |
| | | $h$ | Planck constant | V, F | C, F, P | $\mathcal{U}(1,5)$ | $6.626 \times 10^{-34}$ |
| | | $m$ | Mass | V, F | V, F, P | $\mathcal{U}(1,5)$ | $\mathcal{U}_{\log}(10^{-30}, 10^{-28})$ |
| II.37.1 | $E = \mu(1 + \chi)B$ | $E$ | Energy of magnetic field | V, F | V, F | N/A | N/A |
| | | $\mu$ | Magnetic moment | V, F | V, F | $\mathcal{U}(1,5)$ | $\mathcal{U}_{\log}(10^{-25}, 10^{-23})$ |
| | | $\chi$ | Volume magnetic susceptibility | V, F | V, F | $\mathcal{U}(1,5)$ | $\mathcal{U}_{\log}(10^4, 10^6)$ |
| | | $B$ | Magnetic field strength | V, F | V, F | $\mathcal{U}(1,5)$ | $\mathcal{U}_{\log}(10^{-3}, 10^{-1})$ |

Table S6: Medium set of our proposed datasets (part 4).

| Eq. ID | Formula | Symbols | | Properties | | Distributions | |
|---|---|---|---|---|---|---|---|
| | | | | Original | Ours | Original | Ours |
| III.4.32 | $n = \frac{1}{\exp(h\omega/2\pi kT)-1}$ | $n$ | Average number of photons | V, F | V, F, P | N/A | N/A |
| | | $h$ | Planck constant | V, F | C, F, P | $\mathcal{U}(1,5)$ | $6.626 \times 10^{-34}$ |
| | | $\omega$ | Frequency | V, F | V, F, P | $\mathcal{U}(1,5)$ | $\mathcal{U}_{\log}(10^8, 10^{10})$ |
| | | $k$ | Boltzmann constant | V, F | C, F, P | $\mathcal{U}(1,5)$ | $1.381 \times 10^{-23}$ |
| | | $T$ | Temperature | V, F | V, F, P | $\mathcal{U}(1,5)$ | $\mathcal{U}_{\log}(10^1, 10^3)$ |
| III.8.54 | $|C|^2 = \sin^2\left(\frac{2\pi At}{h}\right)$ | $|C|^2$ | Probability | V, F | V, F, NN | N/A | N/A |
| | | $A$ | Energy | V, F | V, F | $\mathcal{U}(1,2)$ | $\mathcal{U}_{\log}(10^{-18}, 10^{-16})$ |
| | | $t$ | Time | V, F | V, F, NN | $\mathcal{U}(1,2)$ | $\mathcal{U}_{\log}(10^{-18}, 10^{-16})$ |
| | | $h$ | Planck constant | V, F | C, F, P | $\mathcal{U}(1,4)$ | $6.626 \times 10^{-34}$ |
| III.13.18 | $v = \frac{4\pi Ab^2}{h}k$ | $v$ | Speed of the waves | V, F | V, F | N/A | N/A |
| | | $A$ | Energy | V, F | V, F | $\mathcal{U}(1,5)$ | $\mathcal{U}_{\log}(10^{-18}, 10^{-16})$ |
| | | $b$ | Lattice constant | V, F | V, F, P | $\mathcal{U}(1,5)$ | $\mathcal{U}_{\log}(10^{-10}, 10^{-8})$ |
| | | $k$ | Wavenumber | V, F | V, F, P | $\mathcal{U}(1,5)$ | $\mathcal{U}_{\log}(10^{-1}, 10^1)$ |
| | | $h$ | Planck constant | V, F | C, F, P | $\mathcal{U}(1,5)$ | $6.626 \times 10^{-34}$ |
| III.14.14 | $I = I_0\left(\exp\left(q\Delta V/\kappa T\right)-1\right)$ | $I$ | Electric Current | V, F | V, F | N/A | N/A |
| | | $I_0$ | Electric current | V, F | V, F | $\mathcal{U}(1,5)$ | $\mathcal{U}_{\log}(10^{-3}, 10^{-1})$ |
| | | $q$ | Electric charge | V, F | V, F, P | $\mathcal{U}(1,2)$ | $\mathcal{U}_{\log}(10^{-22}, 10^{-20})$ |
| | | $\Delta V$ | Voltage | V, F | V, F | $\mathcal{U}(1,2)$ | $\mathcal{U}_{\log}(10^{-1}, 10^1)$ |
| | | $\kappa$ | Boltzmann constant | V, F | C, F, P | $\mathcal{U}(1,2)$ | $1.381 \times 10^{-23}$ |
| | | $T$ | Temperature | V, F | V, F, P | $\mathcal{U}(1,2)$ | $\mathcal{U}_{\log}(10^1, 10^3)$ |
| III.15.12 | $E = 2A\left(1 - \cos\left(kd\right)\right)$ | $E$ | Energy | V, F | V, F, P | N/A | N/A |
| | | $A$ | Amplitude | V, F | V, F, P | $\mathcal{U}(1,5)$ | $\mathcal{U}_{\log}(10^{-18}, 10^{-16})$ |
| | | $k$ | Propagation coefficient | V, F | V, F, P | $\mathcal{U}(1,5)$ | $\mathcal{U}_{\log}(10^{-1}, 10^1)$ |
| | | $d$ | Lattice constant | V, F | V, F, P | $\mathcal{U}(1,5)$ | $\mathcal{U}_{\log}(10^{-10}, 10^{-8})$ |
| III.15.14 | $m = \frac{h^2}{8\pi^2 Ab^2}$ | $m$ | Effective mass | V, F | V, F, P | N/A | N/A |
| | | $h$ | Planck constant | V, F | C, F, P | $\mathcal{U}(1,5)$ | $6.626 \times 10^{-34}$ |
| | | $A$ | Amplitude | V, F | V, F, P | $\mathcal{U}(1,5)$ | $\mathcal{U}_{\log}(10^{-18}, 10^{-16})$ |
| | | $b$ | Lattice constant | V, F | V, F, P | $\mathcal{U}(1,5)$ | $\mathcal{U}_{\log}(10^{-10}, 10^{-8})$ |
| III.17.37 | $f = \beta(1 + \alpha\cos\theta)$ | $f$ | Distribution | V, F | V, F | N/A | N/A |
| | | $\beta$ | Variable | V, F | V, F, P | $\mathcal{U}(1,5)$ | $\mathcal{U}_{\log}(10^{-18}, 10^{-16})$ |
| | | $\alpha$ | Variable | V, F | V, F | $\mathcal{U}(1,5)$ | $\mathcal{U}_{\log}(10^{-18}, 10^{-16})$ |
| | | $\theta$ | Angle | V, F | V, F, NN | $\mathcal{U}(1,5)$ | $\mathcal{U}(0, 2\pi)$ |
| III.19.51 | $E = -\frac{mq^4}{2(4\pi\epsilon)^2(h/(2\pi))^2 n^2}$ | $E$ | Energy | V, F | V, F, P | N/A | N/A |
| | | $m$ | Mass | V, F | V, F, P | $\mathcal{U}(1,5)$ | $\mathcal{U}_{\log}(10^{-30}, 10^{-28})$ |
| | | $q$ | Electric charge | V, F | V, F | $\mathcal{U}(1,5)$ | $\mathcal{U}_{\log}(10^{-11}, 10^{-9})$ |
| | | $\epsilon$ | Vacuum permittivity | V, F | C, F, P | $\mathcal{U}(1,5)$ | $8.854 \times 10^{-12}$ |
| | | $h$ | Planck constant | V, F | C, F, P | $\mathcal{U}(1,5)$ | $6.626 \times 10^{-34}$ |
| | | $n$ | Number of protons | V, F | V, I, P | $\mathcal{U}(1,5)$ | $\mathcal{U}_{\log}(10^0, 10^2)$ |
| B8 | $U = \frac{E}{1+\frac{E}{mc^2}(1-\cos\theta)}$ | $U$ | Variable | V, F | V, F, P | N/A | N/A |
| | | $E$ | Electromagnetic energy | V, F | V, F, P | $\mathcal{U}(1,3)$ | $\mathcal{U}_{\log}(10^{-24}, 10^{-22})$ |
| | | $m$ | Electron mass | V, F | C, F, P | $\mathcal{U}(1,3)$ | $9.109 \times 10^{-31}$ |
| | | $c$ | Speed of light | V, F | C, F, P | $\mathcal{U}(1,3)$ | $2.998 \times 10^8$ |
| | | $\theta$ | Incidence angle | V, F | V, F | $\mathcal{U}(1,3)$ | $\mathcal{U}(-\pi, \pi)$ |
| B18 | $\rho = \frac{3}{8\pi G}\left(\frac{c^2 k_\mathrm{f}}{a_\mathrm{f}^2} + H^2\right)$ | $\rho$ | Variable | V, F | V, F | N/A | N/A |
| | | $G$ | Gravitational constant | V, F | C, F, P | $\mathcal{U}(1,5)$ | $6.674 \times 10^{-11}$ |
| | | $c$ | Speed of light | V, F | C, F, P | $\mathcal{U}(1,5)$ | $2.998 \times 10^8$ |
| | | $k_\mathrm{f}$ | Variable | V, F | V, F | $\mathcal{U}(1,5)$ | $\mathcal{U}_{\log}(10^1, 10^3)$ |
| | | $a_\mathrm{f}$ | Distance | V, F | V, F, P | $\mathcal{U}(1,5)$ | $\mathcal{U}_{\log}(10^8, 10^{10})$ |
| | | $H$ | Variable | V, F | V, F | $\mathcal{U}(1,5)$ | $\mathcal{U}_{\log}(10^0, 10^2)$ |

Table S7: Hard set of our proposed datasets (part 1).

| Eq. ID | Formula | Symbols | | Properties Original | Ours | Distributions Original | Ours |
|---|---|---|---|---|---|---|---|
| I.6.20 | $f = \exp\left(-\frac{\theta^2}{2\sigma^2}\right)/\sqrt{2\pi\sigma^2}$ | $f$ | Probability density function | V, F | V, F | N/A | N/A |
| | | $\theta$ | Position | V, F | V, F | $\mathcal{U}(1,3)$ | $\mathcal{U}_{\log}(10^{-1}, 10^1)$ |
| | | $\sigma$ | Standard deviation | V, F | V, F, P | $\mathcal{U}(1,3)$ | $\mathcal{U}_{\log}(10^{-1}, 10^1)$ |
| I.6.20a | $f = \exp\left(-\frac{\theta^2}{2}\right)/\sqrt{2\pi}$ | $f$ | Probability density function | V, F | V, F | N/A | N/A |
| | | $\theta$ | Position | V, F | V, F | $\mathcal{U}(1,3)$ | $\mathcal{U}_{\log}(10^{-1}, 10^1)$ |
| I.6.20b | $f = \exp\left(-\frac{(\theta-\theta_1)^2}{2\sigma^2}\right)/\sqrt{2\pi}\sigma$ | $f$ | Probability density function | V, F | V, F | N/A | N/A |
| | | $\theta$ | Position | V, F | V, F | $\mathcal{U}(1,3)$ | $\mathcal{U}_{\log}(10^{-1}, 10^1)$ |
| | | $\theta_1$ | Position | V, F | V, F | $\mathcal{U}(1,3)$ | $\mathcal{U}_{\log}(10^{-1}, 10^1)$ |
| | | $\sigma$ | Standard deviation | V, F | V, F, P | $\mathcal{U}(1,3)$ | $\mathcal{U}_{\log}(10^{-1}, 10^1)$ |
| I.9.18 | $F = \dfrac{Gm_1m_2}{(x_2-x_1)^2 + (y_2-y_1)^2 + (z_2-z_1)^2}$ | $F$ | Force of gravity | V, F | V, F | N/A | N/A |
| | | $G$ | Gravitational constant | V, F | C, F, P | $\mathcal{U}(1,2)$ | $6.674 \times 10^{-11}$ |
| | | $m_1$ | Mass | V, F | V, F, P | $\mathcal{U}(1,2)$ | $\mathcal{U}_{\log}(10^0, 10^3)$ |
| | | $m_2$ | Mass | V, F | V, F, P | $\mathcal{U}(1,2)$ | $\mathcal{U}_{\log}(10^0, 10^3)$ |
| | | $x_2$ | Position | V, F | V, F | $\mathcal{U}(1,2)$ | $\mathcal{U}_{\log}(10^0, 10^1)$ |
| | | $x_1$ | Position | V, F | V, F | $\mathcal{U}(3,4)$ | $\mathcal{U}_{\log}(10^0, 10^1)$ |
| | | $y_2$ | Position | V, F | V, F | $\mathcal{U}(1,2)$ | $\mathcal{U}_{\log}(10^0, 10^1)$ |
| | | $y_1$ | Position | V, F | V, F | $\mathcal{U}(3,4)$ | $\mathcal{U}_{\log}(10^0, 10^1)$ |
| | | $z_2$ | Position | V, F | V, F | $\mathcal{U}(1,2)$ | $\mathcal{U}_{\log}(10^0, 10^1)$ |
| | | $z_1$ | Position | V, F | V, F | $\mathcal{U}(3,4)$ | $\mathcal{U}_{\log}(10^0, 10^1)$ |
| I.15.3t | $t_1 = \dfrac{t - ux/c^2}{\sqrt{1 - u^2/c^2}}$ | $t_1$ | Time | V, F | V, F | N/A | N/A |
| | | $t$ | Time | V, F | V, F, NN | $\mathcal{U}(1,5)$ | $\mathcal{U}_{\log}(10^{-6}, 10^{-4})$ |
| | | $u$ | Velocity | V, F | V, F | $\mathcal{U}(1,2)$ | $\mathcal{U}_{\log}(10^5, 10^7)$ |
| | | $x$ | Position | V, F | V, F | $\mathcal{U}(1,5)$ | $\mathcal{U}_{\log}(10^0, 10^2)$ |
| | | $c$ | Speed of light | V, F | C, F, P | $\mathcal{U}(3,10)$ | $2.998 \times 10^8$ |
| I.15.3x | $x_1 = \dfrac{x - ut}{\sqrt{1 - u^2/c^2}}$ | $x_1$ | Position | V, F | V, F | N/A | N/A |
| | | $x$ | Position | V, F | V, F | $\mathcal{U}(5,10)$ | $\mathcal{U}_{\log}(10^0, 10^2)$ |
| | | $u$ | Velocity | V, F | V, F | $\mathcal{U}(1,2)$ | $\mathcal{U}_{\log}(10^6, 10^8)$ |
| | | $t$ | Time | V, F | V, F | $\mathcal{U}(1,2)$ | $\mathcal{U}_{\log}(10^{-6}, 10^{-4})$ |
| | | $c$ | Speed of light | V, F | C, F, P | $\mathcal{U}(3,20)$ | $2.998 \times 10^8$ |
| I.29.16 | $x = \sqrt{x_1^2 + x_2^2 + 2x_1x_2\cos(\theta_1 - \theta_2)}$ | $x$ | Wavelength | V, F | V, F, P | N/A | N/A |
| | | $x_1$ | Wavelength | V, F | V, F, P | $\mathcal{U}(1,5)$ | $\mathcal{U}_{\log}(10^{-1}, 10^1)$ |
| | | $x_2$ | Wavelength | V, F | V, F, P | $\mathcal{U}(1,5)$ | $\mathcal{U}_{\log}(10^{-1}, 10^1)$ |
| | | $\theta_1$ | Angle | V, F | V, F, NN | $\mathcal{U}(1,5)$ | $\mathcal{U}(0, 2\pi)$ |
| | | $\theta_2$ | Angle | V, F | V, F, NN | $\mathcal{U}(1,5)$ | $\mathcal{U}(0, 2\pi)$ |
| I.30.3 | $I = I_0 \dfrac{\sin^2(n\theta/2)}{\sin^2(\theta/2)}$ | $I$ | Amplitude of combined wave | V, F | V, F | N/A | N/A |
| | | $I_0$ | Amplitude of wave | V, F | V, F, P | $\mathcal{U}(1,5)$ | $\mathcal{U}_{\log}(10^{-3}, 10^{-1})$ |
| | | $n$ | The number of waves | V, F | V, I, P | $\mathcal{U}(1,5)$ | $\mathcal{U}_{\log}(10^1, 10^3)$ |
| | | $\theta$ | Phase difference | V, F | V, F | $\mathcal{U}(1,5)$ | $\mathcal{U}(-2\pi, 2\pi)$ |
| I.32.17 | $P = \left(\frac{1}{2}\epsilon cE^2\right)\left(\frac{8\pi r^2}{3}\right)\left(\frac{\omega^4}{(\omega^2-\omega_0^2)^2}\right)$ | $P$ | Energy | V, F | V, F, P | N/A | N/A |
| | | $\epsilon$ | Vacuum permittivity | V, F | C, F, P | $\mathcal{U}(1,2)$ | $8.854 \times 10^{-12}$ |
| | | $c$ | Speed of light | V, F | C, F, P | $\mathcal{U}(1,2)$ | $2.998 \times 10^8$ |
| | | $E$ | Magnitude of electric field | V, F | V, F, P | $\mathcal{U}(1,2)$ | $\mathcal{U}_{\log}(10^1, 10^3)$ |
| | | $r$ | Radius | V, F | V, F, P | $\mathcal{U}(1,2)$ | $\mathcal{U}_{\log}(10^{-2}, 10^0)$ |
| | | $\omega$ | Frequency of electromagnetic waves | V, F | V, F | $\mathcal{U}(1,2)$ | $\mathcal{U}_{\log}(10^9, 10^{11})$ |
| | | $\omega_0$ | Frequency of electromagnetic waves | V, F | V, F | $\mathcal{U}(3,5)$ | $\mathcal{U}_{\log}(10^9, 10^{11})$ |
| I.34.14 | $\omega = \dfrac{1 + v/c}{\sqrt{1 - v^2/c^2}}\omega_0$ | $\omega$ | Frequency of electromagnetic waves | V, F | V, F | N/A | N/A |
| | | $v$ | Velocity | V, F | V, F | $\mathcal{U}(1,2)$ | $\mathcal{U}_{\log}(10^6, 10^8)$ |
| | | $c$ | Speed of light | V, F | C, F, P | $\mathcal{U}(3,10)$ | $2.998 \times 10^8$ |
| | | $\omega_0$ | Frequency of electromagnetic waves | V, F | V, F, P | $\mathcal{U}(1,5)$ | $\mathcal{U}_{\log}(10^9, 10^{11})$ |

Table S8: Hard set of our proposed datasets (part 2).

| Eq. ID | Formula | Symbols | | Properties Original | Ours | Distributions Original | Ours |
|---|---|---|---|---|---|---|---|
| I.37.4 | $I_{12} = I_1 + I_2$ $+ 2\sqrt{I_1 I_2}\cos\delta$ | $I_{12}$ | Amplitude of wave | V, F | V, F, P | N/A | N/A |
| | | $I_1$ | Amplitude of wave | V, F | V, F, P | $\mathcal{U}(1,5)$ | $\mathcal{U}_{\log}(10^{-1},10^{-3})$ |
| | | $I_2$ | Amplitude of wave | V, F | V, F, P | $\mathcal{U}(1,5)$ | $\mathcal{U}_{\log}(10^{-1},10^{-3})$ |
| | | $\delta$ | Phase difference | V, F | V, F | $\mathcal{U}(1,5)$ | $\mathcal{U}(0,\pi)$ |
| I.39.22 | $P = \frac{nkT}{V}$ | $P$ | Pressure | V, F | V, F, P | N/A | N/A |
| | | $n$ | Number of molecules | V, F | V, I⋆, P | $\mathcal{U}(1,5)$ | $\mathcal{U}_{\log}(10^{23},10^{25})$ |
| | | $k$ | Boltzmann constant | V, F | C, F, P | $\mathcal{U}(1,5)$ | $1.381\times10^{-23}$ |
| | | $T$ | Temperature | V, F | V, F, P | $\mathcal{U}(1,5)$ | $\mathcal{U}_{\log}(10^1,10^3)$ |
| | | $V$ | Volume | V, F | V, F, P | $\mathcal{U}(1,5)$ | $\mathcal{U}_{\log}(10^{-5},10^{-3})$ |
| I.40.1 | $n = n_0\exp\left(-mgx/kT\right)$ | $n$ | Molecular density | V, F | V, F, P | N/A | N/A |
| | | $n_0$ | Molecular density | V, F | V, F, P | $\mathcal{U}(1,5)$ | $\mathcal{U}_{\log}(10^{25},10^{27})$ |
| | | $m$ | Mass | V, F | V, F, P | $\mathcal{U}(1,5)$ | $\mathcal{U}_{\log}(10^{-24},10^{-22})$ |
| | | $g$ | Gravitational acceleration | V, F | C, F, P | $\mathcal{U}(1,5)$ | $9.807\times10^0$ |
| | | $x$ | Height | V, F | V, F, P | $\mathcal{U}(1,5)$ | $\mathcal{U}_{\log}(10^{-2},10^0)$ |
| | | $k$ | Boltzmann constant | V, F | C, F, P | $\mathcal{U}(1,5)$ | $1.381\times10^{-23}$ |
| | | $T$ | Temperature | V, F | V, F, P | $\mathcal{U}(1,5)$ | $\mathcal{U}_{\log}(10^1,10^3)$ |
| I.41.16 | $L_{\mathrm{rad}} = \dfrac{h}{2\pi}$ $\dfrac{\omega^3}{\pi^2 c^2(\exp(h\omega/2\pi kT)-1)}$ | $L_{\mathrm{rad}}$ | Radiation per frequency | V, F | V, F, P | N/A | N/A |
| | | $h$ | Planck constant | V, F | C, F, P | $\mathcal{U}(1,5)$ | $6.626\times10^{-34}$ |
| | | $\omega$ | Frequency of electromagnetic wave | V, F | V, F, P | $\mathcal{U}(1,5)$ | $\mathcal{U}_{\log}(10^{-1},10^1)$ |
| | | $c$ | Speed of light | V, F | C, F, P | $\mathcal{U}(1,5)$ | $2.998\times10^8$ |
| | | $k$ | Boltzmann constant | V, F | C, F, P | $\mathcal{U}(1,5)$ | $1.381\times10^{-23}$ |
| | | $T$ | Temperature | V, F | V, F, P | $\mathcal{U}(1,5)$ | $\mathcal{U}_{\log}(10^1,10^3)$ |
| I.44.4 | $Q = nkT\ln(\frac{V_2}{V_1})$ | $Q$ | Energy | V, F | V, F | N/A | N/A |
| | | $n$ | Number of molecules | V, F | V, I⋆, P | $\mathcal{U}(1,5)$ | $\mathcal{U}_{\log}(10^{23},10^{25})$ |
| | | $k$ | Boltzmann constant | V, F | C, F, P | $\mathcal{U}(1,5)$ | $1.381\times10^{-23}$ |
| | | $T$ | Temperature | V, F | V, F, P | $\mathcal{U}(1,5)$ | $\mathcal{U}_{\log}(10^1,10^3)$ |
| | | $V_2$ | Volume | V, F | V, F, P | $\mathcal{U}(1,5)$ | $\mathcal{U}_{\log}(10^{-5},10^{-3})$ |
| | | $V_1$ | Volume | V, F | V, F, P | $\mathcal{U}(1,5)$ | $\mathcal{U}_{\log}(10^{-5},10^{-3})$ |
| I.50.26 | $x = K\left(\cos\omega t + \epsilon\cos^2\omega t\right)$ | $x$ | Amplitude | V, F | V, F | N/A | N/A |
| | | $K$ | Amplitude | V, F | V, F, P | $\mathcal{U}(1,3)$ | $\mathcal{U}_{\log}(10^{-1},10^1)$ |
| | | $\omega$ | Angular velocity | V, F | V, F | $\mathcal{U}(1,3)$ | $\mathcal{U}_{\log}(10^1,10^3)$ |
| | | $t$ | Time | V, F | V, F, NN | $\mathcal{U}(1,3)$ | $\mathcal{U}_{\log}(10^{-3},10^{-1})$ |
| | | $\epsilon$ | Variable | V, F | V, F | $\mathcal{U}(1,3)$ | $\mathcal{U}_{\log}(10^{-3},10^{-1})$ |
| II.6.15a | $E = \frac{p}{4\pi\epsilon}\frac{3z}{r^5}\sqrt{x^2+y^2}$ | $E$ | Electric field | V, F | V, F | N/A | N/A |
| | | $p$ | Electric dipole moment | V, F | V, F | $\mathcal{U}(1,3)$ | $\mathcal{U}_{\log}(10^{-22},10^{-20})$ |
| | | $\epsilon$ | Vacuum permittivity | V, F | C, F, P | $\mathcal{U}(1,3)$ | $8.854\times10^{-12}$ |
| | | $z$ | Position | V, F | V, F | $\mathcal{U}(1,3)$ | $\mathcal{U}_{\log}(10^{-10},10^{-8})$ |
| | | $r$ | Distance | V, F | V, F, P | $\mathcal{U}(1,3)$ | $\mathcal{U}_{\log}(10^{-10},10^{-8})$ |
| | | $x$ | Position | V, F | V, F | $\mathcal{U}(1,3)$ | $\mathcal{U}_{\log}(10^{-10},10^{-8})$ |
| | | $y$ | Position | V, F | V, F | $\mathcal{U}(1,3)$ | $\mathcal{U}_{\log}(10^{-10},10^{-8})$ |
| II.6.15b | $E = \frac{p}{4\pi\epsilon}\frac{3\cos\theta\sin\theta}{r^3}$ | $E$ | Electric field | V, F | V, F | N/A | N/A |
| | | $p$ | Electric dipole moment | V, F | V, F | $\mathcal{U}(1,3)$ | $\mathcal{U}_{\log}(10^{-22},10^{-20})$ |
| | | $\epsilon$ | Vacuum permittivity | V, F | C, F, P | $\mathcal{U}(1,3)$ | $8.854\times10^{-12}$ |
| | | $\theta$ | Angle | V, F | V, F | $\mathcal{U}(1,3)$ | $\mathcal{U}(0,\pi)$ |
| | | $r$ | Distance | V, F | V, F, P | $\mathcal{U}(1,3)$ | $\mathcal{U}_{\log}(10^{-10},10^{-8})$ |
| II.11.17 | $n = n_0\left(1+\frac{p_0 E\cos\theta}{kT}\right)$ | $n$ | Number of polar molecules per angle per unit volume | V, F | V, F | N/A | N/A |
| | | $n_0$ | Number of molecules per unit volume | V, F | V, F, P | $\mathcal{U}(1,3)$ | $\mathcal{U}_{\log}(10^{27},10^{29})$ |
| | | $p_0$ | Electric dipole moment | V, F | V, F | $\mathcal{U}(1,3)$ | $\mathcal{U}_{\log}(10^{-22},10^{-20})$ |
| | | $E$ | Magnitude of electric field | V, F | V, F | $\mathcal{U}(1,3)$ | $\mathcal{U}_{\log}(10^1,10^3)$ |
| | | $\theta$ | Angle | V, F | V, F, NN | $\mathcal{U}(1,3)$ | $\mathcal{U}(0,2\pi)$ |
| | | $k$ | Boltzmann constant | V, F | C, F, P | $\mathcal{U}(1,3)$ | $1.381\times10^{-23}$ |
| | | $T$ | Temperature | V, F | V, F, P | $\mathcal{U}(1,3)$ | $\mathcal{U}_{\log}(10^1,10^3)$ |
| II.11.20 | $P = \frac{n_0 p_0^2 E}{3kT}$ | $P$ | Polarizability | V, F | V, F | N/A | N/A |
| | | $n_0$ | Number of atom | V, F | V, I⋆, P | $\mathcal{U}(1,5)$ | $\mathcal{U}_{\log}(10^{23},10^{25})$ |
| | | $p_0$ | Electric dipole moment | V, F | V, F | $\mathcal{U}(1,5)$ | $\mathcal{U}_{\log}(10^{-22},10^{-20})$ |
| | | $E$ | Magnitude of electric field | V, F | V, F | $\mathcal{U}(1,5)$ | $\mathcal{U}_{\log}(10^1,10^3)$ |
| | | $k$ | Boltzmann constant | V, F | C, F, P | $\mathcal{U}(1,5)$ | $1.381\times10^{-23}$ |
| | | $T$ | Temperature | V, F | V, F, P | $\mathcal{U}(1,5)$ | $\mathcal{U}_{\log}(10^1,10^3)$ |

Table S9: Hard set of our proposed datasets (part 3).

| Eq. ID | Formula | Symbols | | Properties | | Distributions | |
|---|---|---|---|---|---|---|---|
| | | | | Original | Ours | Original | Ours |
| II.11.27 | $P = \frac{N\alpha}{1-(n\alpha/3)}\epsilon E$ | $P$ | Polarizability | V, F | V, F | N/A | N/A |
| | | $N$ | Number of atom | V, F | V, I⋆, P | $\mathcal{U}(0,1)$ | $\mathcal{U}_{\log}(10^{23},10^{25})$ |
| | | $\alpha$ | Molecular polarizability | V, F | V, F, P | $\mathcal{U}(0,1)$ | $\mathcal{U}_{\log}(10^{-33},10^{-31})$ |
| | | $\epsilon$ | Vacuum permittivity | V, F | C, F, P | $\mathcal{U}(1,2)$ | $8.854 \times 10^{-12}$ |
| | | $E$ | Magnitude of electric field | V, F | V, F, P | $\mathcal{U}(1,2)$ | $\mathcal{U}_{\log}(10^{1},10^{3})$ |
| II.11.28 | $\kappa = 1 + \frac{N\alpha}{1-(N\alpha/3)}$ | $\kappa$ | Electric dipole moment per unit volume | V, F | V, F | N/A | N/A |
| | | $N$ | Number of electric dipoles | V, F | V, I⋆, P | $\mathcal{U}(0,1)$ | $\mathcal{U}_{\log}(10^{23},10^{25})$ |
| | | $\alpha$ | Molecular polarizability | V, F | V, F, P | $\mathcal{U}(0,1)$ | $\mathcal{U}_{\log}(10^{-33},10^{-31})$ |
| II.13.23 | $\rho = \frac{\rho_0}{\sqrt{1-v^2/c^2}}$ | $\rho$ | Electric charge density | V, F | V, F, P | N/A | N/A |
| | | $\rho_0$ | Electric charge density | V, F | V, F, P | $\mathcal{U}(1,5)$ | $\mathcal{U}_{\log}(10^{27},10^{29})$ |
| | | $v$ | Velocity | V, F | V, F, P | $\mathcal{U}(1,2)$ | $\mathcal{U}_{\log}(10^{6},10^{8})$ |
| | | $c$ | Speed of light | V, F | C, F, P | $\mathcal{U}(3,10)$ | $2.998 \times 10^8$ |
| II.13.34 | $j = \frac{\rho_0 v}{\sqrt{1-v^2/c^2}}$ | $j$ | Electric current | V, F | V, F | N/A | N/A |
| | | $\rho_0$ | Electric charge density | V, F | V, F, P | $\mathcal{U}(1,5)$ | $\mathcal{U}_{\log}(10^{27},10^{29})$ |
| | | $v$ | Velocity | V, F | V, F, P | $\mathcal{U}(1,2)$ | $\mathcal{U}_{\log}(10^{6},10^{8})$ |
| | | $c$ | Speed of light | V, F | C, F, P | $\mathcal{U}(3,10)$ | $2.998 \times 10^8$ |
| II.24.17 | $k = \sqrt{\omega^2/c^2 - \pi^2/a^2}$ | $k$ | Wavenumber | V, F | V, F, P | N/A | N/A |
| | | $\omega$ | Angular velocity | V, F | V, F | $\mathcal{U}(4,6)$ | $\mathcal{U}_{\log}(10^{9},10^{11})$ |
| | | $c$ | Speed of light | V, F | C, F, P | $\mathcal{U}(1,2)$ | $2.998 \times 10^8$ |
| | | $a$ | Length | V, F | V, F, P | $\mathcal{U}(2,4)$ | $\mathcal{U}_{\log}(10^{-3},10^{-1})$ |
| II.35.18 | $a = \frac{N}{\exp(\mu B/kT) + \exp(-\mu B/kT)}$ | $a$ | Number of atoms with the equivalent magnetic moment | V, F | V, I⋆, P | N/A | N/A |
| | | $N$ | Number of atoms per unit volume | V, F | V, I⋆, P | $\mathcal{U}(1,3)$ | $\mathcal{U}_{\log}(10^{23},10^{25})$ |
| | | $\mu$ | Magnetic moment | V, F | V, F, P | $\mathcal{U}(1,3)$ | $\mathcal{U}_{\log}(10^{-25},10^{-23})$ |
| | | $B$ | Magnetic flux density | V, F | V, F, P | $\mathcal{U}(1,3)$ | $\mathcal{U}_{\log}(10^{-3},10^{-1})$ |
| | | $k$ | Boltzmann constant | V, F | C, F, P | $\mathcal{U}(1,3)$ | $1.381 \times 10^{-23}$ |
| | | $T$ | Temperature | V, F | V, F, P | $\mathcal{U}(1,3)$ | $\mathcal{U}_{\log}(10^{1},10^{3})$ |
| II.35.21 | $M = N\mu \tanh\left(\frac{\mu B}{kT}\right)$ | $M$ | Number of magnetized atoms | V, F | V, I⋆, P | N/A | N/A |
| | | $N$ | Number of atom | V, F | V, I⋆, P | $\mathcal{U}(1,5)$ | $\mathcal{U}_{\log}(10^{23},10^{25})$ |
| | | $\mu$ | Magnetic moment | V, F | V, F, P | $\mathcal{U}(1,5)$ | $\mathcal{U}_{\log}(10^{-25},10^{-23})$ |
| | | $B$ | Magnetic flux density | V, F | V, F, P | $\mathcal{U}(1,5)$ | $\mathcal{U}_{\log}(10^{-3},10^{-1})$ |
| | | $k$ | Boltzmann constant | V, F | C, F, P | $\mathcal{U}(1,5)$ | $1.381 \times 10^{-23}$ |
| | | $T$ | Temperature | V, F | V, F, P | $\mathcal{U}(1,5)$ | $\mathcal{U}_{\log}(10^{1},10^{3})$ |
| II.36.38 | $x = \frac{\mu H}{kT} + \frac{\mu \lambda}{\epsilon c^2 kT} M$ | $x$ | Parameter of magnetization | V, F | V, F | N/A | N/A |
| | | $\mu$ | Magnetic moment | V, F | V, F | $\mathcal{U}(1,3)$ | $\mathcal{U}_{\log}(10^{-25},10^{-23})$ |
| | | $H$ | Magnetic field strength | V, F | V, F | $\mathcal{U}(1,3)$ | $\mathcal{U}_{\log}(10^{-3},10^{-1})$ |
| | | $k$ | Boltzmann constant | V, F | C, F, P | $\mathcal{U}(1,3)$ | $1.381 \times 10^{-23}$ |
| | | $T$ | Temperature | V, F | V, F, P | $\mathcal{U}(1,3)$ | $\mathcal{U}_{\log}(10^{1},10^{3})$ |
| | | $\lambda$ | Constant | V, F | V, F, NN | $\mathcal{U}(1,3)$ | $\mathcal{U}(0,1)$ |
| | | $\epsilon$ | Vacuum permittivity | V, F | C, F, P | $\mathcal{U}(1,3)$ | $8.854 \times 10^{-12}$ |
| | | $c$ | Speed of light | V, F | C, F, P | $\mathcal{U}(1,3)$ | $2.998 \times 10^8$ |
| | | $M$ | Number of magnetized atoms | V, F | V, I⋆, P | $\mathcal{U}(1,3)$ | $\mathcal{U}_{\log}(10^{23},10^{25})$ |
| III.4.33 | $E = \frac{h\omega}{2\pi(\exp(h\omega/2\pi kT)-1)}$ | $E$ | Energy | V, F | V, F, P | N/A | N/A |
| | | $h$ | Planck constant | V, F | C, F, P | $\mathcal{U}(1,5)$ | $6.626 \times 10^{-34}$ |
| | | $\omega$ | Frequency | V, F | V, F, P | $\mathcal{U}(1,5)$ | $\mathcal{U}_{\log}(10^{8},10^{10})$ |
| | | $k$ | Boltzmann constant | V, F | C, F, P | $\mathcal{U}(1,5)$ | $1.381 \times 10^{-23}$ |
| | | $T$ | Temperature | V, F | V, F, P | $\mathcal{U}(1,5)$ | $\mathcal{U}_{\log}(10^{1},10^{3})$ |
| III.9.52 | $P_{\text{I}\to\text{II}} = \left(\frac{2\pi\mu E t}{h}\right)^2 \frac{\sin^2((\omega-\omega_0)t/2)}{(\omega-\omega_0)t/2)^2}$ | $P_{\text{I}\to\text{II}}$ | Probability | V, F | V, F, NN | N/A | N/A |
| | | $\mu$ | Electric dipole moment | V, F | V, F | $\mathcal{U}(1,3)$ | $\mathcal{U}_{\log}(10^{-22},10^{-20})$ |
| | | $E$ | Magnitude of electric field | V, F | V, F | $\mathcal{U}(1,3)$ | $\mathcal{U}_{\log}(10^{1},10^{3})$ |
| | | $t$ | Time | V, F | V, F, NN | $\mathcal{U}(1,3)$ | $\mathcal{U}_{\log}(10^{-18},10^{-16})$ |
| | | $h$ | Planck constant | V, F | C, F, P | $\mathcal{U}(1,3)$ | $6.626 \times 10^{-34}$ |
| | | $\omega$ | Frequency | V, F | V, F, P | $\mathcal{U}(1,5)$ | $\mathcal{U}_{\log}(10^{8},10^{10})$ |
| | | $\omega_0$ | Resonant frequency | V, F | V, F, P | $\mathcal{U}(1,5)$ | $\mathcal{U}_{\log}(10^{8},10^{10})$ |

Table S10: Hard set of our proposed datasets (part 4).

| Eq. ID | Formula | Symbol | Description | Properties Original | Properties Ours | Distributions Original | Distributions Ours |
|---|---|---|---|---|---|---|---|
| III.10.19 | $E = \mu\sqrt{B_x^2 + B_y^2 + B_z^2}$ | $E$ | Energy | V, F | V, F | N/A | N/A |
| | | $\mu$ | Magnetic moment | V, F | V, F | $\mathcal{U}(1,5)$ | $\mathcal{U}_{\log}(10^{-25}, 10^{-23})$ |
| | | $B_x$ | Element of magnetic field | V, F | V, F | $\mathcal{U}(1,5)$ | $\mathcal{U}_{\log}(10^{-3}, 10^{-1})$ |
| | | $B_y$ | Element of magnetic field | V, F | V, F | $\mathcal{U}(1,5)$ | $\mathcal{U}_{\log}(10^{-3}, 10^{-1})$ |
| | | $B_z$ | Element of magnetic field | V, F | V, F | $\mathcal{U}(1,5)$ | $\mathcal{U}_{\log}(10^{-3}, 10^{-1})$ |
| III.21.20 | $J = -\rho\frac{q}{m}A$ | $J$ | Electric Current | V, F | V, F | N/A | N/A |
| | | $\rho$ | Electric charge density | V, F | V, F, N | $\mathcal{U}(1,5)$ | $\mathcal{U}_{\log}(10^{27}, 10^{29})$ |
| | | $q$ | Electric charge | V, F | V, F, N | $\mathcal{U}(1,5)$ | $\mathcal{U}_{\log}(10^{-11}, 10^{-9})$ |
| | | $A$ | Magnetic vector potential | V, F | V, F | $\mathcal{U}(1,5)$ | $\mathcal{U}_{\log}(10^{-3}, 10^{-1})$ |
| | | $m$ | Mass | V, F | V, F, P | $\mathcal{U}(1,5)$ | $\mathcal{U}_{\log}(10^{-30}, 10^{-28})$ |
| B1 | $A = \left(\frac{Z_1 Z_2 \alpha h c}{4E\sin^2(\theta/2)}\right)^2$ | $A$ | Differential scattering cross section | V, F | V, F | N/A | N/A |
| | | $Z_1$ | Atomic number | V, F | V, I, P | $\mathcal{U}(1,2)$ | $\mathcal{U}_{\log}(10^0, 10^1)$ |
| | | $Z_2$ | Atomic number | V, F | V, I, P | $\mathcal{U}(1,2)$ | $\mathcal{U}_{\log}(10^0, 10^1)$ |
| | | $\alpha$ | Fine structure constant | V, F | C, F, P | $\mathcal{U}(1,5)$ | $7.297 \times 10^{-3}$ |
| | | $h$ | Dirac's constant | V, F | C, F, P | $\mathcal{U}(1,2)$ | $1.055 \times 10^{-34}$ |
| | | $c$ | Speed of light | V, F | C, F, P | $\mathcal{U}(1,2)$ | $2.998 \times 10^8$ |
| | | $E$ | Non-relativistic kinetic energy | V, F | V, F, P | $\mathcal{U}(1,3)$ | $\mathcal{U}_{\log}(10^{-18}, 10^{-16})$ |
| | | $\theta$ | Scattering angle | V, F | V, F, NN | $\mathcal{U}(1,3)$ | $\mathcal{U}(0, 2\pi)$ |
| B2 | $k = \frac{mk_G}{L^2}\left(1+\sqrt{1+\frac{2EL^2}{mk_G^2}}\cos(\theta_1-\theta_2)\right)$ | $k$ | Variable | V, F | V, F | N/A | N/A |
| | | $m$ | Mass (The Earth) | V, F | V, F, P | $\mathcal{U}(1,3)$ | $\mathcal{U}_{\log}(10^{23}, 10^{25})$ |
| | | $k_G$ | Variable | V, F | V, F, P | $\mathcal{U}(1,3)$ | $\mathcal{U}_{\log}(10^9, 10^{11})$ |
| | | $L$ | Distance | V, F | V, F, P | $\mathcal{U}(1,3)$ | $\mathcal{U}_{\log}(10^8, 10^{10})$ |
| | | $E$ | Energy | V, F | V, F, P | $\mathcal{U}(1,3)$ | $\mathcal{U}_{\log}(10^{25}, 10^{27})$ |
| | | $\theta_1$ | Angle | V, F | V, F, NN | $\mathcal{U}(0,6)$ | $\mathcal{U}(0, 2\pi)$ |
| | | $\theta_2$ | Angle | V, F | V, F, NN | $\mathcal{U}(0,6)$ | $\mathcal{U}(0, 2\pi)$ |
| B3 | $r = \frac{d(1-\alpha^2)}{1+\alpha\cos(\theta_1-\theta_2)}$ | $r$ | Distance | V, F | V, F, P | N/A | N/A |
| | | $d$ | Semimajor axis of elliptical orbit | V, F | V, F, P | $\mathcal{U}(1,3)$ | $\mathcal{U}_{\log}(10^8, 10^{10})$ |
| | | $\alpha$ | Orbital eccentricity | V, F | V, F, P | $\mathcal{U}(2,4)$ | $\mathcal{U}(0,1)$ |
| | | $\theta_1$ | Angle | V, F | V, F, NN | $\mathcal{U}(4,5)$ | $\mathcal{U}(0, 2\pi)$ |
| | | $\theta_2$ | Angle | V, F | V, F, NN | $\mathcal{U}(4,5)$ | $\mathcal{U}(0, 2\pi)$ |
| B4 | $v = \sqrt{\frac{2}{m}\left(E-U-\frac{L^2}{2mr^2}\right)}$ | $v$ | Velocity | V, F | V, F, P | N/A | N/A |
| | | $m$ | Mass (The Earth) | V, F | V, F, P | $\mathcal{U}(1,3)$ | $\mathcal{U}_{\log}(10^{23}, 10^{25})$ |
| | | $E$ | Energy | V, F | V, F, P | $\mathcal{U}(8,12)$ | $\mathcal{U}_{\log}(10^{25}, 10^{27})$ |
| | | $U$ | Potential energy | V, F | V, F, P | $\mathcal{U}(1,3)$ | $\mathcal{U}_{\log}(10^{25}, 10^{27})$ |
| | | $L$ | Angular momentum | V, F | V, F | $\mathcal{U}(1,3)$ | $\mathcal{U}_{\log}(10^8, 10^{10})$ |
| | | $r$ | Distance | V, F | V, F, P | $\mathcal{U}(1,3)$ | $\mathcal{U}_{\log}(10^8, 10^{10})$ |
| B5 | $t = \frac{2\pi d^{3/2}}{\sqrt{G(m_1+m_2)}}$ | $t$ | Orbital period | V, F | V, F, P | N/A | N/A |
| | | $d$ | Semimajor axis of elliptical orbit | V, F | V, F, P | $\mathcal{U}(1,3)$ | $\mathcal{U}_{\log}(10^8, 10^{10})$ |
| | | $G$ | Gravitational constant | V, F | C, F, P | $\mathcal{U}(1,3)$ | $6.674 \times 10^{-11}$ |
| | | $m_1$ | Mass (The Earth) | V, F | V, F, P | $\mathcal{U}(1,3)$ | $\mathcal{U}_{\log}(10^{23}, 10^{25})$ |
| | | $m_2$ | Mass (The Earth) | V, F | V, F, P | $\mathcal{U}(1,3)$ | $\mathcal{U}_{\log}(10^{23}, 10^{25})$ |
| B6 | $\alpha = \sqrt{1+\frac{2\epsilon^2 EL^2}{m(Z_1 Z_2 q^2)^2}}$ | $\alpha$ | Orbital eccentricity | V, F | V, F, P | N/A | N/A |
| | | $\epsilon$ | Energy | V, F | V, F | $\mathcal{U}(1,3)$ | $\mathcal{U}_{\log}(10^{-18}, 10^{-16})$ |
| | | $E$ | Energy | V, F | V, F, P | $\mathcal{U}(1,3)$ | $\mathcal{U}_{\log}(10^{-18}, 10^{-16})$ |
| | | $L$ | Distance | V, F | V, F, P | $\mathcal{U}(1,3)$ | $\mathcal{U}_{\log}(10^{-10}, 10^{-8})$ |
| | | $m$ | Mass | V, F | V, F, P | $\mathcal{U}(1,3)$ | $\mathcal{U}_{\log}(10^{-30}, 10^{-28})$ |
| | | $Z_1$ | Atomic number | V, F | V, I, P | $\mathcal{U}(1,3)$ | $\mathcal{U}_{\log}(10^0, 10^1)$ |
| | | $Z_2$ | Atomic number | V, F | V, I, P | $\mathcal{U}(1,3)$ | $\mathcal{U}_{\log}(10^0, 10^1)$ |
| | | $q$ | Electric charge | V, F | V, F | $\mathcal{U}(1,3)$ | $\mathcal{U}_{\log}(10^{-11}, 10^{-9})$ |
| B7 | $H = \sqrt{\frac{8\pi G\rho}{3} - \frac{k_f c^2}{a_f^2}}$ | $H$ | Hubble's constant | V, F | V, F, P | N/A | N/A |
| | | $G$ | Gravitational constant | V, F | C, F, P | $\mathcal{U}(1,3)$ | $6.674 \times 10^{-11}$ |
| | | $\rho$ | Density of the Universe | V, F | V, F, P | $\mathcal{U}(1,3)$ | $\mathcal{U}_{\log}(10^{-27}, 10^{-25})$ |
| | | $k_f$ | Spacetime curvature | V, F | V, I | $\mathcal{U}(1,2)$ | $\mathcal{U}(-1, 1)$ |
| | | $c$ | Speed of light | V, F | C, F, P | $\mathcal{U}(1,2)$ | $2.998 \times 10^8$ |
| | | $a_f$ | Radius | V, F | V, F, P | $\mathcal{U}(1,3)$ | $\mathcal{U}_{\log}(10^8, 10^{10})$ |
| B9 | $P = -\frac{32}{5}\frac{G^4}{c^5}\frac{(m_1 m_2)^2(m_1+m_2)}{r^5}$ | $P$ | Gravitational wave energy | V, F | V, F | N/A | N/A |
| | | $G$ | Gravitational constant | V, F | C, F, P | $\mathcal{U}(1,2)$ | $6.674 \times 10^{-11}$ |
| | | $c$ | Speed of light | V, F | C, F, P | $\mathcal{U}(1,2)$ | $2.998 \times 10^8$ |
| | | $m_1$ | Mass | V, F | V, F, P | $\mathcal{U}(1,5)$ | $\mathcal{U}_{\log}(10^{23}, 10^{25})$ |
| | | $m_2$ | Mass | V, F | V, F, P | $\mathcal{U}(1,5)$ | $\mathcal{U}_{\log}(10^{23}, 10^{25})$ |
| | | $r$ | Distance | V, F | V, F, P | $\mathcal{U}(1,2)$ | $\mathcal{U}_{\log}(10^8, 10^{10})$ |

Table S11: Hard set of our proposed datasets (part 5).

| Eq. ID | Formula | Symbols | | Properties Original | Ours | Distributions Original | Ours |
|---|---|---|---|---|---|---|---|
| B10 | $\cos\theta_1 = \frac{\cos\theta_2 - v/c}{(1-v/c)\cos\theta_2}$ | $\cos\theta_1$ | Value | V, F | V, F | N/A | N/A |
| | | $\theta_2$ | Angle | V, F | V, F | $\mathcal{U}(1,3)$ | $\mathcal{U}(0,2\pi)$ |
| | | $v$ | Velocity | V, F | V, F | $\mathcal{U}(1,3)$ | $\mathcal{U}_{\log}(10^5,10^7)$ |
| | | $c$ | Speed of light | V, F | C, F, P | $\mathcal{U}(4,6)$ | $2.998\times10^8$ |
| B11 | $I = I_0\left(\frac{\sin(\alpha/2)}{\alpha/2}\frac{\sin(N\delta/2)}{\sin(\delta/2)}\right)^2$ | $I$ | Wave intensity | V, F | V, F, P | N/A | N/A |
| | | $I_0$ | Amplitude of wave | V, F | V, F, P | $\mathcal{U}(1,3)$ | $\mathcal{U}_{\log}(10^{-3},10^{-1})$ |
| | | $\alpha$ | Wavelength of X-ray | V, F | V, F, P | $\mathcal{U}(1,3)$ | $\mathcal{U}_{\log}(10^{-11},10^{-9})$ |
| | | $N$ | Number of phase difference | V, F | V, I,P | $\mathcal{U}(1,2)$ | $\mathcal{U}_{\log}(10^0,10^2)$ |
| | | $\delta$ | Wavelength of X-ray | V, F | V, F, P | $\mathcal{U}(1,3)$ | $\mathcal{U}_{\log}(10^{-11},10^{-9})$ |
| B12 | $F = \frac{q}{4\pi\epsilon y^2}\left(4\pi\epsilon V_e d - \frac{qdy^3}{(y^2-d^2)^2}\right)$ | $F$ | Force | V, F | V, F | N/A | N/A |
| | | $q$ | Electric charge | V, F | V, F | $\mathcal{U}(1,5)$ | $\mathcal{U}_{\log}(10^{-3},10^{-1})$ |
| | | $\epsilon$ | Vacuum permittivity | V, F | C, F, P | $\mathcal{U}(1,5)$ | $8.854\times10^{-12}$ |
| | | $y$ | Distance | V, F | V, F, P | $\mathcal{U}(1,3)$ | $\mathcal{U}_{\log}(10^{-2},10^0)$ |
| | | $V_e$ | Voltage | V, F | V, F | $\mathcal{U}(1,5)$ | $\mathcal{U}_{\log}(10^{-1},10^1)$ |
| | | $d$ | Distance | V, F | V, F, P | $\mathcal{U}(4,6)$ | $\mathcal{U}_{\log}(10^{-2},10^0)$ |
| B13 | $V_e = \frac{q}{4\pi\epsilon\sqrt{r^2+d^2-2dr\cos\alpha}}$ | $V_e$ | Potential | V, F | V, F | N/A | N/A |
| | | $\epsilon$ | permittivity | V, F | V, F, P | $\mathcal{U}(1,5)$ | $\mathcal{U}_{\log}(10^{-12},10^{-10})$ |
| | | $q$ | Electric charge | V, F | V, F | $\mathcal{U}(1,5)$ | $\mathcal{U}_{\log}(10^{-3},10^{-1})$ |
| | | $r$ | Distance | V, F | V, F, P | $\mathcal{U}(1,3)$ | $\mathcal{U}_{\log}(10^{-2},10^0)$ |
| | | $d$ | Distance between dipoles | V, F | V, F, P | $\mathcal{U}(4,6)$ | $\mathcal{U}_{\log}(10^{-2},10^0)$ |
| | | $\alpha$ | Angle | V, F | V, F | $\mathcal{U}(0,6)$ | $\mathcal{U}(0,2\pi)$ |
| B14 | $V_e = E_f\cos\theta\left(\frac{\alpha-1}{\alpha+2}\frac{d^3}{r^2} - r\right)$ | $V_e$ | Potential (out) | V, F | V, F | N/A | N/A |
| | | $E_f$ | Magnitude of electric field | V, F | V, F | $\mathcal{U}(1,5)$ | $\mathcal{U}_{\log}(10^1,10^3)$ |
| | | $\theta$ | Angle | V, F | V, F | $\mathcal{U}(0,6)$ | $\mathcal{U}(0,2\pi)$ |
| | | $r$ | Distance | V, F | V, F, P | $\mathcal{U}(1,5)$ | $\mathcal{U}_{\log}(10^{-2},10^0)$ |
| | | $d$ | Radius of dielectric sphere | V, F | V, F, P | $\mathcal{U}(1,5)$ | $\mathcal{U}_{\log}(10^{-2},10^0)$ |
| | | $\alpha$ | Polarizability | V, F | V, F | $\mathcal{U}(1,5)$ | $\mathcal{U}_{\log}(10^{-1},10^1)$ |
| B15 | $\omega_0 = \frac{\sqrt{1-\frac{v^2}{c^2}}}{1+\frac{v}{c}\cos\theta}\omega$ | $\omega_0$ | Frequency of electromagnetic waves | V, F | V, F | N/A | N/A |
| | | $v$ | Velocity | V, F | V, F, P | $\mathcal{U}(1,3)$ | $\mathcal{U}_{\log}(10^5,10^7)$ |
| | | $c$ | Speed of light | V, F | C, F, P | $\mathcal{U}(5,20)$ | $2.998\times10^8$ |
| | | $\omega$ | Frequency of electromagnetic waves | V, F | V, F, P | $\mathcal{U}(1,5)$ | $\mathcal{U}_{\log}(10^9,10^{11})$ |
| | | $\theta$ | Angle | V, F | V, F | $\mathcal{U}(0,6)$ | $\mathcal{U}(0,2\pi)$ |
| B16 | $E = qV_e + \sqrt{(p-qA)^2c^2+m^2c^4}$ | $E$ | Energy | V, F | V, F | N/A | N/A |
| | | $p$ | Momentum | V, F | V, F | $\mathcal{U}(1,5)$ | $\mathcal{U}_{\log}(10^{-9},10^{-7})$ |
| | | $q$ | Electric charge | V, F | V, F | $\mathcal{U}(1,5)$ | $\mathcal{U}_{\log}(10^{-11},10^{-9})$ |
| | | $A$ | Vector potential | V, F | V, F | $\mathcal{U}(1,5)$ | $\mathcal{U}_{\log}(10^1,10^3)$ |
| | | $c$ | Speed of light | V, F | C, F, P | $\mathcal{U}(1,5)$ | $2.998\times10^8$ |
| | | $m$ | Mass | V, F | V, F, P | $\mathcal{U}(1,5)$ | $\mathcal{U}_{\log}(10^{-30},10^{-28})$ |
| | | $V_e$ | Voltage | V, F | V, F | $\mathcal{U}(1,5)$ | $\mathcal{U}_{\log}(10^{-1},10^1)$ |
| B17 | $E = \frac{1}{2m}\left(p^2+m^2\omega^2x^2\left(1+\alpha\frac{x}{y}\right)\right)$ | $E$ | Energy | V, F | V, F | N/A | N/A |
| | | $m$ | Mass | V, F | V, F, P | $\mathcal{U}(1,5)$ | $\mathcal{U}_{\log}(10^{-30},10^{-28})$ |
| | | $p$ | Momentum | V, F | V, F | $\mathcal{U}(1,5)$ | $\mathcal{U}_{\log}(10^{-9},10^{-7})$ |
| | | $\omega$ | Frequency of electromagnetic waves | V, F | V, F, P | $\mathcal{U}(1,5)$ | $\mathcal{U}_{\log}(10^9,10^{11})$ |
| | | $x$ | Position | V, F | V, F | $\mathcal{U}(1,5)$ | $\mathcal{U}_{\log}(10^{-11},10^{-9})$ |
| | | $\alpha$ | Deviation from the harmonic oscillator | V, F | V, F | $\mathcal{U}(1,5)$ | $\mathcal{U}_{\log}(10^{-1},10^1)$ |
| | | $y$ | Distance | V, F | V, F, P | $\mathcal{U}(1,5)$ | $\mathcal{U}_{\log}(10^{-11},10^{-9})$ |
| B19 | $p_f = -\frac{1}{8\pi G}\left(\frac{c^4 k_f}{a_f^2} + c^2 H^2(1-2\alpha)\right)$ | $p_f$ | Pressure | V, F | V, F | N/A | N/A |
| | | $G$ | Gravitational constant | V, F | C, F, P | $\mathcal{U}(1,5)$ | $6.674\times10^{-11}$ |
| | | $c$ | Speed of light | V, F | C, F, P | $\mathcal{U}(1,5)$ | $2.998\times10^8$ |
| | | $k_f$ | Variable | V, F | V, F | $\mathcal{U}(1,5)$ | $\mathcal{U}_{\log}(10^1,10^3)$ |
| | | $a_f$ | Distance | V, F | V, F, P | $\mathcal{U}(1,5)$ | $\mathcal{U}_{\log}(10^8,10^{10})$ |
| | | $H$ | Variable | V, F | V, F, P | $\mathcal{U}(1,5)$ | $\mathcal{U}_{\log}(10^0,10^2)$ |
| | | $\alpha$ | Variable | V, F | V, F | $\mathcal{U}(1,5)$ | $\mathcal{U}(-10,10)$ |
| B20 | $A = \frac{\alpha^2 h^2}{4\pi m^2 c^2}\left(\frac{\omega_0}{\omega}\right)^2\left(\frac{\omega_0}{\omega}+\frac{\omega}{\omega_0}-\sin^2\theta\right)$ | $A$ | Differential cross section | V, F | V, F | N/A | N/A |
| | | $\alpha$ | Fine structure constant | V, F | C, F, P | $\mathcal{U}(1,5)$ | $7.297\times10^{-3}$ |
| | | $h$ | Planck constant | V, F | C, F, P | $\mathcal{U}(1,5)$ | $6.626\times10^{-34}$ |
| | | $m$ | Electron mass | V, F | C, F, P | $\mathcal{U}(1,5)$ | $9.109\times10^{-31}$ |
| | | $c$ | Speed of light | V, F | C, F, P | $\mathcal{U}(1,5)$ | $2.998\times10^8$ |
| | | $\omega_0$ | Frequency | V, F | V, F, P | $\mathcal{U}(1,5)$ | $\mathcal{U}_{\log}(10^9,10^{11})$ |
| | | $\omega$ | Frequency | V, F | V, F, P | $\mathcal{U}(1,5)$ | $\mathcal{U}_{\log}(10^9,10^{11})$ |
| | | $\theta$ | Scattering angle | V, F | V, F | $\mathcal{U}(0,6)$ | $\mathcal{U}(0,2\pi)$ |

# B    HYPERPARAMETERS FOR FIVE EXISTING SR BASELINES

Table S12 shows the hyperparameter space for the five existing symbolic regression baselines. The hyperparameters of gplearn (Koza & Poli, 2005) [6], AFP (Schmidt & Lipson, 2011), and AFP-FE (Schmidt & Lipson, 2009) [7] are optimized by Optuna (Akiba et al., 2019), a hyperparameter optimization framework.

Table S12: Hyperparameter sets for the five existing symbolic regression baselines.

| Method | Hyperparameter sets |
|---|---|
| gplearn | 100 trials with random combinations of the following hyperparameter spaces:
*population_size*: $\mathcal{U}(10^2, 10^3)$, *generations*: $\mathcal{U}(10, 10^2)$,
*stopping_criteria*: $\mathcal{U}(10^{-10}, 10^{-2})$, *warm_start*: {True, False},
*const_range*: {None, $(-1.0, 1.0), (-10, 10), (-10^2, 10^2), (-10^3, 10^3), (-10^4, 10^4)$},
*max_samples*: $\mathcal{U}(0.9, 1.0)$, *parsimony_coefficient*: $\mathcal{U}(10^{-3}, 10^{-2})$ |
| AFP | 100 trials with random combinations of the following hyperparameter spaces:
*popsize*: $\mathcal{U}(100, 1000)$, *g*: $\mathcal{U}(250, 2500)$, *stop_threshold*: $\mathcal{U}(10^{-10}, 10^{-2})$,
*op_list*: {['n', 'v', '+', '-', '*', '/', 'exp', 'log', '2', '3', 'sqrt'],
['n', 'v', '+', '-', '*', '/', 'exp', 'log', '2', '3', 'sqrt', 'sin', 'cos']} |
| AFP-FE | 100 trials with random combinations of the following hyperparameter spaces:
*popsize*: $\mathcal{U}(100, 1000)$, *g*: $\mathcal{U}(250, 2500)$, *stop_threshold*: $\mathcal{U}(10^{-10}, 10^{-2})$,
*op_list*: {['n', 'v', '+', '-', '*', '/', 'exp', 'log', '2', '3', 'sqrt'],
['n', 'v', '+', '-', '*', '/', 'exp', 'log', '2', '3', 'sqrt', 'sin', 'cos']} |
| AI Feynman | {*bftt*: 60, *epoch*: 300, *op*: '7ops.txt', *poly_deg*: 3},
{*bftt*: 60, *epoch*: 300, *op*: '10ops.txt', *poly_deg*: 3},
{*bftt*: 60, *epoch*: 300, *op*: '14ops.txt', *poly_deg*: 3},
{*bftt*: 60, *epoch*: 300, *op*: '19ops.txt', *poly_deg*: 3},
{*bftt*: 120, *epoch*: 300, *op*: '14ops.txt', *poly_deg*: 4},
{*bftt*: 120, *epoch*: 300, *op*: '19ops.txt', *poly_deg*: 4},
{*bftt*: 60, *epoch*: 500, *op*: '7ops.txt', *poly_deg*: 3},
{*bftt*: 60, *epoch*: 500, *op*: '10ops.txt', *poly_deg*: 3},
{*bftt*: 60, *epoch*: 500, *op*: '14ops.txt', *poly_deg*: 3},
{*bftt*: 60, *epoch*: 500, *op*: '19ops.txt', *poly_deg*: 3} |
| DSR | {*seed*: 1, *function_set*: ['add', 'sub', 'mul', 'div', 'sin', 'cos', 'exp', 'log']},
{*seed*: 2, *function_set*: ['add', 'sub', 'mul', 'div', 'sin', 'cos', 'exp', 'log']},
{*seed*: 3, *function_set*: ['add', 'sub', 'mul', 'div', 'sin', 'cos', 'exp', 'log']},
{*seed*: 4, *function_set*: ['add', 'sub', 'mul', 'div', 'sin', 'cos', 'exp', 'log']},
{*seed*: 5, *function_set*: ['add', 'sub', 'mul', 'div', 'sin', 'cos', 'exp', 'log']},
{*seed*: 1, *function_set*: ['add', 'sub', 'mul', 'div', 'sin', 'cos', 'exp', 'log', 'const']},
{*seed*: 2, *function_set*: ['add', 'sub', 'mul', 'div', 'sin', 'cos', 'exp', 'log', 'const']},
{*seed*: 3, *function_set*: ['add', 'sub', 'mul', 'div', 'sin', 'cos', 'exp', 'log', 'const']},
{*seed*: 4, *function_set*: ['add', 'sub', 'mul', 'div', 'sin', 'cos', 'exp', 'log', 'const']},
{*seed*: 5, *function_set*: ['add', 'sub', 'mul', 'div', 'sin', 'cos', 'exp', 'log', 'const']} |

[6] https://gplearn.readthedocs.io/en/stable/reference.html#symbolic-regressor
[7] https://github.com/cavalab/ellyn

## C    Symbolic Transformer

In this section, we describe our Symbolic Transformer baseline and pretraining strategy, and discuss the degree of the test data leakage during the pretraining. We repeat that while the Symbolic Transformer itself is a new model, the main contribution of this work lies in the datasets and benchmark of symbolic regression for scientific discovery (SRSD). We also note that our Symbolic Transformer is not necessarily designed to show improvements over the existing Transformer-based symbolic regression models (Valipour et al., 2021; Biggio et al., 2021) including contemporary work such as Kamienny et al. (2022). Our Transformer-based baseline is simply inspired by recent advances in deep learning, specifically Transformer-based high-performance, modern, and flexible models such as Vaswani et al. (2017); Devlin et al. (2019); Dosovitskiy et al. (2020).

### C.1    Architecture Design

A Symbolic Transformer is an encoder-decoder network to predict a binary equation tree corresponding to input tabular data points. The encoder inputs sampled tabular data from an equation and outputs variable-wise features. The decoder inputs them and outputs a preorder sequence of tokens in an autoregressive manner to build a binary equation tree.

First, we describe an encoder that inputs tabular data and outputs variable-wise features. The tabular data here have the following properties:

- Each row is a data point sampled from the expected equation
- Each column corresponds to the input variable of the equation ($x_1$, $x_2$, $\cdots$) or the output of the equation ($y$).

For such data, the relation between the input tabular data and the output equation is desired to have the following properties:

- Permutation-invariant in each row; since there is no meaningful order between each sampled point, the output equation is expected not to change no matter how the rows are rearranged.

- Permutation-equivariant for each column and variable; when the columns of the input tabular data are rearranged, the order of variables in the output equation is expected to change corresponding to the input column. (*e.g.*, assume the formula $y = x_1 - x_2$, if the $x_1$ column and the $x_2$ column of the input tabular data are swapped, the expected output is $y = x_2 - x_1$.)

To handle such permutation-invariant and permutation-equivariant data structures, the ideas of PointNet (Charles et al., 2017), which handles 3D point clouds, and Deep Sets (Zaheer et al., 2017), which handles set data, can be referred to. The approach proposed in these papers is to utilize point-wise transformation using Shared Multi Layer Perceptron (MLP) (permutation-equivariant operation) and aggregation using Pooling (permutation-invariant operation).

When applied to tabular data, a sample point-wise feature can be obtained by combining Shared MLP and column-wise (variable-wise) pooling. Similarly, a variable-wise feature can be obtained by combining Shared MLP and row-wise (sample point-wise) Pooling. In both operations, by aggregating once and then combining the features with the features of each cell (this idea is used in semantic segmentation network in PointNet (Charles et al., 2017)), it can be expected that the whole information of aggregated direction will propagate to each cell. We call this operation feature splatting. Note that it is a permutation-equivariant operation.

The proposed network architecture is shown in Fig. S1. The proposed encoder consists of two encoding blocks, each block consisting of 1.) Shared MLP, 2.) row-wise (sample point-wise) pooling, 3.) row-wise feature splatting, 4.) Shared MLP, 5.) column-wise (variable-wise) pooling, and 6.) column-wise feature splatting. The encoder is permutation-equivariant, and information of each cell propagates to both the row-axis and column-axis. Finally, row-wise (sample point-wise) pooling can obtain the variable-wise feature.

The decoder of our Symbolic Transformer is a Transformer decoder (Vaswani et al., 2017) that takes a variable-wise feature and a preorder sequence of tokens (to express the equation as a binary equation tree) as input and outputs the next token. The next token is estimated with Multi-head Attention

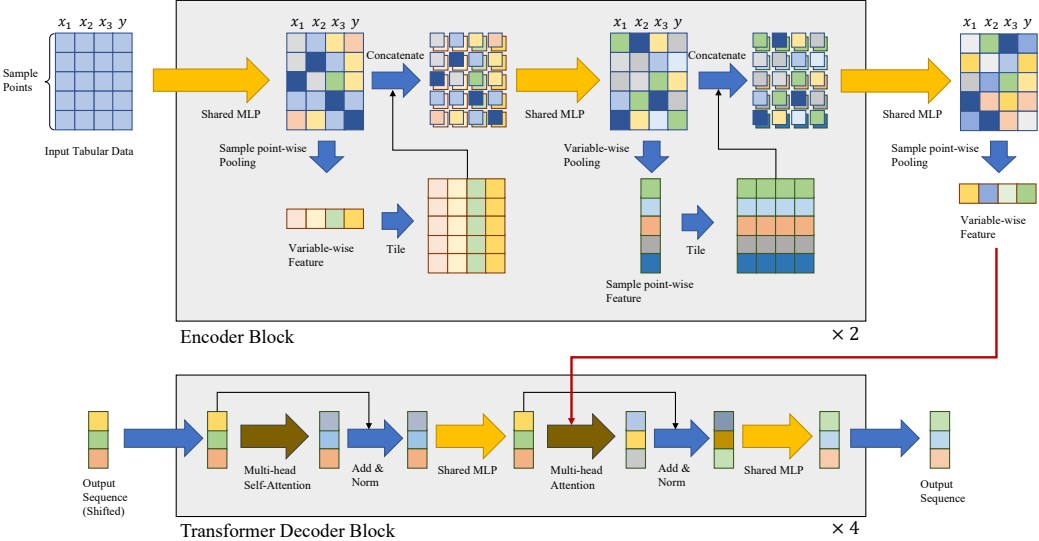

Figure S1: Architecture of Symbolic Transformer.

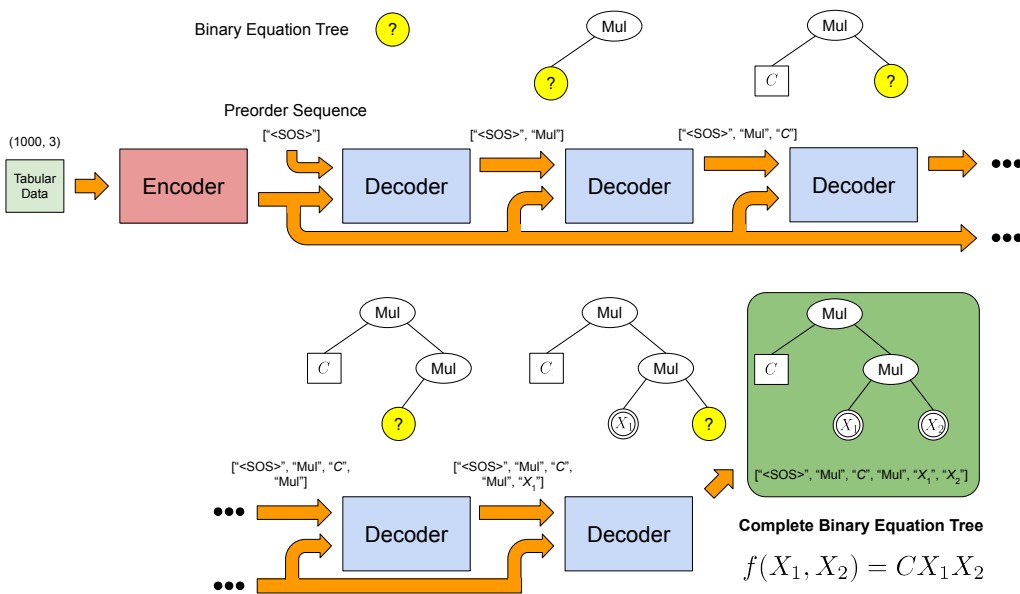

Figure S2: Illustration of Symbolic Transformer's inference process for given tabular data which consists of 1,000 samples. Each sample is a 3-dimensional vector whose first two and last values correspond to two variables and one target value, respectively. Yellow nodes indicate next tokens (nodes) to be predicted.

over the variable-wise feature from the input tokens. During training, we use the teacher forcing method (Williams & Zipser, 1989) and feed the first $(j-1)$ tokens in the ground-truth preorder sequence to the decoder for predicting the next ($j$-th) token and computing cross-entropy loss with a scheduled sampling strategy (Liu et al., 2021a). During inference, the decoder predicts the next token autoregressively, using a sequence of previously-predicted tokens as input tokens as illustrated in Fig. S2. From the resulting preorder sequence of tokens, the binary equation tree is generated as the output of the Symbolic Transformer method.

For each training batch, we also mask next token candidates as the Symbolic Transformer model will know the potential maximum number of variables in the target equation from the shape of the training batch. Suppose that a training batch consists of tabular data whose shape is (1,000, 3) [8], the decoder should not predict three or more unique variable tokens as part of the preorder sequence.

It should be worth noting that when predicting the next token of a preorder sequence, we take advantage of properties of binary equation tree for efficient inference. Since we predict a preorder sequence to build a binary equation tree, we can terminate the inference of the next token when all the child nodes in the binary equation tree built from the resulting sequence expresses are tokens for either constant or variable symbols *i.e.*, the binary equation tree is complete and cannot be extend anymore, thus we can terminate the inference without predicting "<EOS>", the end of sequence token used in standard seq2seq tasks (Sutskever et al., 2014).

## C.2    PRETRAINING AND HYPERPARAMETERS

Here, we describe our pretraining strategy for the Symbolic Transformer baseline. As described in Section C.1], the decoder of the Symbolic Transformer is trained to predict a next token of a preorder sequence to express the true binary equation tree, and thus requires both 1) a true sequence of the tokens and 2) tabular data (samples).

The 120 SRSD datasets proposed in this study, however, should not be directly used for pretraining the Symbolic Transformer baseline. Since the datasets are designed to assess the potential of symbolic regression methods towards scientific discovery, we cannot use the SRSD datasets for both pretraining and testing the Symbolic Transformer baseline. Otherwise, it will result in having the baseline method learn from the pairs of tabular data and true sequences during training session and test the method using tabular data from a similar distribution, which may be considered as test data leakage.

Moreover, pretraining datasets are usually significantly larger than the target datasets (which are the SRSD datasets in this study) (Devlin et al., 2019; Liu et al., 2019; Dosovitskiy et al., 2020). To address the issues, we generated a large-scale set of synthesized datasets for pretraining the Symbolic Transformer baseline. Specifically, we built a bi-gram Naïve Bayes language model to generate preorder sequences of tokens that express binary equation trees like those in the SRSD datasets. Having randomly generated a target equation for pretraining, we randomly chose $k$ for each of the variables in the equation independently to define its sampling range $\mathcal{U}_{\log}(10^{k-1}, 10^{k+1})$. Given a generated equation, we created up to 10 sets of tabular datasets with different sampling ranges we randomly chose as described above.

Using the approach, we generated 24,232 synthesized datasets. In the datasets, there are 4,501 generated equations, and 1,841 of them are unique sequences. We pretrained the Symbolic Transformer baseline on the synthesized datasets for three epochs, using the SGD optimizer and a linear learning rate scheduler with warmup (1,000 steps) [9] where the initial learning rate and weight decay are 0.1 and $10^{-4}$, respectively. A training batch at each step consists of 1,000 random samples from one of the synthesized datasets, and one epoch in this pretraining consists of 24,232 training steps. The Symbolic Transformer is implemented with PyTorch (Paszke et al., 2019), and we implemented the pretraining and evaluation sessions, using torchdistill (Matsubara, 2021) for securing reproducibility of the experiments.

## C.3    ARE THE TEST DATASETS LEAKED WHILE PRETRAINING SYMBOLIC TRANSFORMER?

In terms of normalized edit distance we proposed, the Symbolic Transformer baseline performed the best among our baseline methods for the SRSD datasets as shown in Section 5.3. To achieve the results, the Symbolic Transformer baseline model took advantage of being pretrained on a large-scale set of synthesized datasets we generated (See Section C.2). From the performance gap between the Symbolic Transformer and the other baselines, it is reasonable to suspect that the test set of our SRSD datasets may be leaked during the pretraining step. Given a pair of a synthesized training dataset $X_{\mathrm{R}}$ and an SRSD test dataset $X_{\mathrm{S}}$, we assess the degree of the leakage, following the two steps:

---

[8]The last (rightmost) column in input tabular data indicates the target values, thus should not be a input variable for the equation to be predicted.

[9]`https://huggingface.co/docs/transformers/main_classes/optimizer_schedules#transformers.get_linear_schedule_with_warmup`

Step 1  Compute normalized edit distance (Eq. (3)) between the true equations in the two datasets.

Step 2  If the resulting normalized edit distance (Step 1) is zero, then compute intersection of union (IoU) between sampling domains of the datasets by Eq. (S1) for each input variable:

$$\text{IoU}(X_{\text{R}}^i, X_{\text{S}}^i) = \frac{\max\left(\min\left(X_{\text{R}}^i\right), \min\left(X_{\text{S}}^i\right)\right) - \min\left(\max\left(X_{\text{R}}^i\right), \max\left(X_{\text{S}}^i\right)\right)}{\min\left(\min\left(X_{\text{R}}^i\right), \min\left(X_{\text{S}}^i\right)\right) - \max\left(\max\left(X_{\text{R}}^i\right), \max\left(X_{\text{S}}^i\right)\right)}, \quad \text{(S1)}$$

where $X_{\text{R}}^i$ and $X_{\text{S}}^i$ are sets of sampled values in the $i$-th variable for the given synthesized training dataset and the SRSD test dataset, respectively.

Following the procedure above, we assess the degree of the leakage for all the combinations of 120 SRSD datasets versus 24,232 synthesized datasets for pretraining. We take equation-wise average of the IoU values and then compute the average IoU over the 120 equation-wise average IoU values. The resulting mean IoU is 0.00215, and the IoU defined in Eq. (S1) is designed to be normalized in range of 0 to 1. From the result, we conclude that there is no significant leakage from our SRSD datasets while pretraining the Symbolic Transformer.

## D  QUALITATIVE ANALYSIS

This section discusses qualitative analysis for the experimental results above.

Here we highlight some examples that the Symbolic Transformer baseline method perform better with respect to the normalized edit distance than other baselines and vice versa. Taking I.12.4 in Table 1 as an example, $E = \frac{q_1}{4\pi\epsilon r^2}$ is simplified by sympy[10] and converted to a skeleton equation as $f(x) = c_1 \cdot x_1 \cdot x_2^{c_2}$, where $c$'s are constant tokens [11] and $x_1$ and $x_2$ are the first and second variables, respectively.

Table S13 shows the predicted skeleton equations and normalized edit distance from the target skeleton. For the specific true symbolic expression, gplean, AFP, and AFP-FE could not finish the training to produce symbolic expressions for the corresponding tabular data within the session timeout, which is part of our runtime constraints. While the skeleton equation produced by the Symbolic Transformer (ST) is not a perfect solution, it is structurally closer to the target skeleton than those produced by AI Feynman and DSR. It seems that both AI Feynman and DSR attempted to fit the training data and minimize their regression errors, but they resulted in overcomplex symbolic expressions.

Table S14 shows a different example (I.14.3 from Table 1) where some of the other baseline methods performed better than the Symbolic Transformer. While DSR and Symbolic Transformer produced overcomplex or simpler solutions, AFP, AFP-FE, and AI Feynman produced a perfect skeleton with respect to normalized edit distance from the true equation. Overall, our Symbolic Transformer baseline seems to prefer simpler symbolic expressions, that may avoid overcomplex symbolic expressions as a result.

## E  INJECTING NOISE TO TARGET VARIABLES

Following SRBench (La Cava et al., 2021), we introduce Gaussian noise with a parameter of noise level $\gamma$ to the target variables in our SRSD datasets. We inject Gaussian noise to each of the datasets separately, following Eq. (S2):

$$y_j^{\text{noise}} = f_{\text{true}}\left(X_j\right) + \epsilon, \quad \epsilon \sim \mathcal{N}\left(0, \gamma\sqrt{\frac{1}{N}\sum_{k=1}^{N} f_{\text{true}}\left(X_k\right)}\right), \quad \text{(S2)}$$

where $1 \leq j \leq N$ and $N$ indicates the number of samples in the dataset.

---

[10]$E(q_1, r) = \frac{q_1}{4\pi\epsilon r^2} \rightarrow f(x) = 8987742437.98822 \cdot x_1/x_2^2$ by substituting the constant values in Table 1.

[11]When computing normalized edit distance, we treat all the constants ($c$'s) in a skeleton equation as duplicate nodes. *i.e.*, As tree nodes, constant indices are not important since those will be estimated separately at the end, and in general the estimated constant values should not affect the edit distance.

Table S13: An example (I.12.4) that Symbolic Transformer performed better than other baselines.

| Target Skeleton | $c_1 \cdot x_1/x_2^{c_2}$ | NED |
|---|---|---|
| AI Feynman | $\tan(x_2/\sqrt{x_2^{c_1}} + c_2)$ | 1.00 |
| DSR | $x_1 \cdot (x_1 + c_1 \cdot \exp((c_2 \cdot \cos(x_2 + c_3) + c_4)/x_2)) \cdot \exp(-x_2)$ | 1.00 |
| ST | $c_1 \cdot x_2^{c_2}$ | 0.167 |

Table S14: An example (I.14.3) that AFP, AFP-FE, and AI Feynman outperformed ST.

| Target Skeleton | $c_1 \cdot x_1 \cdot x_2$ | NED |
|---|---|---|
| AI Feynman[†] | $c_1 \cdot x_1 \cdot x_2$ | 0.00 |
| DSR | $x_1 \cdot x_2 \cdot (c_1 - (c_2 \cdot x_2 + c_3 \cdot \log(\cos(x_2))) \cdot (-x_1 + x_2 + c_4)/x_2)$ | 1.00 |
| ST | $x_1 \cdot x_2$ | 0.250 |

[†]AFT and AFP-FE produced exactly the same skeleton with AI Feynman, which resulted in NED = 0.

Table S15: Normalized edit distances of baselines for noise-injected SRSD (Easy) datasets with different noise levels.

| Noise Level ($\gamma$) \ Method | gplearn | AFP | AFP-FE | AI Feynman | DSR | ST |
|---|---|---|---|---|---|---|
| 0 | 0.876 | 0.703 | 0.712 | 0.646 | 0.551 | **0.435** |
| $10^{-3}$ | 0.928 | 0.799 | 0.814 | 0.797 | 0.820 | **0.435** |
| $10^{-2}$ | 0.940 | 0.824 | 0.880 | 0.870 | 0.793 | **0.435** |
| $10^{-1}$ | 0.948 | 0.823 | 0.960 | 0.882 | 0.841 | **0.461** |

Table S16: Solution rates of common baselines for FSRD and SRSD (Easy, Medium, Hard) datasets.

| Dataset \ Method | gplearn | AFP | AFP-FE | AI Feynman | DSR |
|---|---|---|---|---|---|
| FSRD (Udrescu & Tegmark, 2020) | 15.5% | 20.48% | 26.23% | **52.65%** | 19.71% |
| SRSD (Ours) | 1.67% | 5.83% | 5.83% | 9.17% | **15.0%** |

Table S15 shows normalized edit distances of our baselines for noise-injected SRSD (Easy), reusing the set of noise levels in SRBench (La Cava et al., 2021) *i.e.*, $\gamma \in \{0, 10^{-3}, 10^{-2}, 10^{-1}\}$. Interestingly, our Symbolic Transformer (ST) baseline seems less sensitive to noise injected to target variables than other baseline methods. Overall, the more the injected noise is, the more difficult it would be for the baseline models to (re-)discover the physics law in the data.

## F    SOLUTION RATE COMPARISON - FSRD VS. SRSD -

Table S16 compares the solution rates of the five common baselines for the FSRD and our SRSD datasets. We can confirm that the overall solution rates for our SRSD are significantly degraded compared to those for the FSRD reported in SRBench (La Cava et al., 2021). The results indicate that our SRSD datasets are more challenging than the FSRD in terms of solution rate.

## G    USER STUDY - $R^2$ SCORE VS. NORMALIZED EDIT DISTANCE -

To investigate how aligned with human judges the existing SR and new SRSD evaluation metrics are, we recruited 23 people from industry and academia who either have doctoral degrees (scientists, professors, engineers) or are doctoral students, and performed a user study with approval from an ethics review board. The recruited people are in diverse research fields such as computer science, mathematics, physics, chemistry, material science, aerospace engineering, engineering, medical nutrition, and computational biology. Given a pair of true and estimate equations for an SRSD problem, the subjects were asked to assess an estimated equation on a discretized 1-to-5 scale, where

Table S17: Pearson correlation coefficients (PCCs) between the human judges and SR/SRSD metrics.

| Metrics | PCC | P-value |
|---|---|---|
| $R^2$ score | $4.66 \times 10^{-3}$ | $0.913$ |
| NED | $-0.416$ | $1.85 \times 10^{-24}$ |

1 and 5 indicate "1: Completely different from the true equation" and "5: Equivalent to the true equation" respectively. We chose SRSD problems among the 120 SRSD datasets such that we can obtain from the experimental results in Section 5 at least two different equations estimated by different methods that are best in terms of $R^2$ score and normalized edit distance. There were 24 resulting SRSD problems for the user study.

Table S17 shows Pearson correlation coefficients (PCCs) between the human judges and SR/SRSD evaluation metrics. For normalized edit distance (NED), the Pearson correlation coefficient and p-value were $-0.416$ and $1.85 \times 10^{-24}$ respectively, which show a much stronger and statistically more significant correlation between NED and human judges than one for $R^2$ scores. In other words, results of the user study indicate that normalized edit distance is more aligned with human judges than $R^2$ score, and thus can be a better estimate about how close to the true equations the estimated equations are, in a more human-understandable way.

