# OpenReview forum: "Rethinking Symbolic Regression Datasets and Benchmarks for Scientific Discovery"
_ICLR.cc/2023/Conference — Submitted to ICLR 2023_

### Official Review · Reviewer_CRUL · 2022-10-17

**Confidence:** 3
**Correctness:** 3
**Technical Novelty And Significance:** 3
**Empirical Novelty And Significance:** 2
**Recommendation:** 6

**Clarity, Quality, Novelty And Reproducibility:**

The proposed SRSD datasets expand the FSRD database and appear novel and potentially useful to the scientific discovery via symbolic regression community, however the clarity of the paper could be improved somewhat.

**Strength And Weaknesses:**

Strengths
---
The proposed SRSD datasets address issues present in the FSRD database in order to provide datasets that better align with the goals of symbolic regression for scientific discovery.

The proposed SRSD contains 120 datasets, partitioned into three different sets of varying difficulty level.

The proposed metric of normalized edit distance allows for a more fine-grained analysis into the structural differences between the predicted and true equation models.

The paper is relatively well-written and easy to read and follow.

Weaknesses
---
It is not clear to me how much insight is gained using the normalized edit distance (NED) over existing metrics measuring the success of symbolic regression models. The results suggest that the transformer-based model achieves slightly better NED over existing methods, but it is difficult to judge if this improvement is meaningful from a scientific discovery point of view. Even if two equation models are mostly structurally similar (as measured by NED), they may differ by one or two elements that could perceivably result in a very different equation model not representative of the actual ground truth equation.

The experimental results are not very thorough, and only provide results for a small number of metrics for a small number of baselines. Due to the difficulty evaluating the merit of equation-based scientific discovery, perhaps a metric based on manual inspection with domain experts could give some insight into whether or not the proposed metric (and existing metrics) are well-aligned with how humans judge how close a generated equation is to representing an actual scientific equation.

When proposing a new dataset, the authors should consider documenting their dataset thoroughly using, for instance, "Datasheets for Datasets" (Gebru et al. 2021) or a dataset nutrition label (Holland et al. 2018).

Minor Weaknesses
---
Consider using "\citet" when using citations as a noun.

The solution rate metric is not clearly described.

Why is accuracy and NED not shown in the comparison between the FSRD and SRSD dataset (Table S16)?

**Summary Of The Paper:**

The paper propose a new symbolic regression (SR) dataset for scientific discovery (SRSD) that is a modification of the Feynman symbolic regression database (FSRD) that fixes the following issues present in FSRD: 1) no physical meaning for a number of the datasets, 2) an oversimplified sampling process, 3) distinct equations that are marked as duplicates, and 4) inappropriate or incorrect formulas. The proposed SRSD datasets treats constants (e.g., lightspeed) as constants, samples values for variables with appropriate ranges using experimental observations, and partitions the 120 datasets into easy, medium, and hard sets based on formula operation length and variable domain range. The authors also propose using normalized edit distance to assess the similarity of the structures between the predicted equation models and their respective ground-truth models.

The authors evaluate 5 established and well-known SR methods and one transformer-based SR model on the proposed SRSD datasets and measure performance using the proposed normalized edit distance, accuracy (which captures instances where the predicted and true model differ by a constant or scalar), and solution rate. Overall, the authors find the transformer-based approach outperforms all other methods on normalized edit distance, but performs worse than all other methods in terms of accuracy and solution rate; this suggests the transformer-based approach provides more structurally similar equation models than existing methods.

**Summary Of The Review:**

Although the paper proposes a potentially valuable collection of SRSD datasets and a new evaluation metric measuring normalized edit distance, the dataset is not well-documented and it is unclear whether the new evaluation metric provides significantly meaningful insights in terms of scientific discovery compared to existing evaluation metrics.

---

> ### Author Response · Authors · 2022-11-07
> **1st round Author Response to Reviewer CRUL (1 of 2)**
>
> We thank Reviewer CRUL for recognizing key strengths of our work and their insightful comments. To address one of their concerns, we are currently performing a user study with approval from an ethics review board (we will post a follow-up comment later). We address other concerns in our point-to-point responses below.
>
> ---
>
> ### [R_CRUL-1st_Round-A1]
>
> > Strengths
>
> > The proposed SRSD datasets address issues present in the FSRD database in order to provide datasets that better align with the goals of symbolic regression for scientific discovery.
>
> > The proposed SRSD contains 120 datasets, partitioned into three different sets of varying difficulty level.
>
> > The proposed metric of normalized edit distance allows for a more fine-grained analysis into the structural differences between the predicted and true equation models.
>
> > The paper is relatively well-written and easy to read and follow.
>
> We thank the reviewer for many positive comments and agreeing to key contributions of this work.
>
> ---
>
> ### [R_CRUL-1st_Round-A2]
>
> > Weaknesses
> > It is not clear to me how much insight is gained using the normalized edit distance (NED) over existing metrics measuring the success of symbolic regression models. The results suggest that the transformer-based model achieves slightly better NED over existing methods, but it is difficult to judge if this improvement is meaningful from a scientific discovery point of view. Even if two equation models are mostly structurally similar (as measured by NED), they may differ by one or two elements that could perceivably result in a very different equation model not representative of the actual ground truth equation.
>
> We agree with the reviewer. It would be important to compare the NED to existing metrics and understand how meaningful the NED is.
>
> To discuss it, we are recruiting people from industry and academia and performing a user study with approval from an ethics review board. We will make a separate follow-up comment to share the updates.
>
>
>
> ---
>
> ### [R_CRUL-1st_Round-A3]
>
> > The experimental results are not very thorough, and only provide results for a small number of metrics for a small number of baselines. Due to the difficulty evaluating the merit of equation-based scientific discovery, perhaps a metric based on manual inspection with domain experts could give some insight into whether or not the proposed metric (and existing metrics) are well-aligned with how humans judge how close a generated equation is to representing an actual scientific equation.
>
> We respectfully disagree with the reviewer about the assessments on experimental results, numbers of metrics and baselines.
>
> 1. As explained at the beginning of Section 3.2.2, there will be non-trivial training cost required to train a symbolic regression model for each of the 120 SRSD datasets since 120 separate training sessions to assess the symbolic regression approach.
> 2. Each of the 120 separate training sessions (each pair of SR problem and SR method) includes a hyperparameter tuning process that takes up to 100 training trials (See Section B).
> 3. Due to the significant amount of training cost, we chose the top-5 symbolic regression methods in terms of solution late for the FRSD datasets tested in the SRBench study.
> 4. We discuss the performance of SR methods using three metrics: $R^2$-driven accuracy, solution rate, and NED. The first two metrics are used in the SRBench study as well. In appendix (Sections D and E), we also performed a qualitative analysis and discussed performance when injecting noise to target variables.
>
> For the human-based evaluation, see our answer in [R_CRUL-1st_Round-A2].
>
>
> ---
>
> ### [R_CRUL-1st_Round-A4]
>
> > When proposing a new dataset, the authors should consider documenting their dataset thoroughly using, for instance, "Datasheets for Datasets" (Gebru et al. 2021) or a dataset nutrition label (Holland et al. 2018).
>
> We thank the reviewer for suggestions. For documenting our datasets and code repository, we followed [Hugging Face Dataset Card Creation Guide](https://github.com/huggingface/datasets/blob/main/templates/README_guide.md), which is also [aligned with “Datasheets for Datasets”](https://github.com/huggingface/datasets/blob/main/CONTRIBUTING.md#how-to-contribute-to-the-dataset-cards).
>
>
> ---
>
> (continued below)

---

> > ### Author Response · Authors · 2022-11-07
> > **1st round Author Response to Reviewer CRUL (2 of 2)**
> >
> > (continued from above)
> >
> > ---
> >
> > ### [R_CRUL-1st_Round-A5]
> >
> > > Minor Weaknesses
> >
> > > Consider using "\citet" when using citations as a noun.
> >
> > We thank the reviewer for the suggestion. We found such cases and replaced \citep with \citet for them.
> >
> > ---
> >
> > ### [R_CRUL-1st_Round-A6]
> >
> > > The solution rate metric is not clearly described.
> >
> > It is introduced in the SRBench study and shows a percentage of the estimated symbolic regression models that match the true models (equations) as explained at the end of Section 2.
> > Suppose two symbolic expressions: $f$ (true) and $g$ (estimated).
> > If either conditions 1 or 2 is true, the estimated symbolic expression $g$ is treated as correct for the solution rate.
> > 1. $f - g = a$
> > 2. $f / g = b$
> > for some constant values $a$ and $b$.
> >
> > Case 1: $f = x + 1$ and $g = x^2$
> >
> > Either conditions 1 or 2 is not true, then $g$ is treated as incorrect for the solution rate.
> >
> > Case 2: $f = x + 1$ and $g = x - 100$
> >
> > Condition 1 is true ($f - g = -99$, which is constant), then $g$ is treated as correct for the solution rate.
> >
> > We also clarified the solution rate at footnote 3 in the revision.
> >
> > ---
> >
> > ### [R_CRUL-1st_Round-A7]
> >
> > > Why is accuracy and NED not shown in the comparison between the FSRD and SRSD dataset (Table S16)?
> >
> > As pointed out in Section 3.1 (Oversimplified sampling process), the FRSD datasets do not distinguish between constants and variables (e.g., speed of light is treated as a variable and randomly sampled in range of 1 to 5). Thus, equation trees (symbolic expressions as trees) to be estimated from the FRSD datasets are different from those to be estimated from the SRSD datasets, which makes it difficult to discuss NED comparison between the FSRD and SRSD datasets.
> >
> > ---
> >
> > ### [R_CRUL-1st_Round-A8]
> >
> > > The proposed SRSD datasets expand the FSRD database and appear novel and potentially useful to the scientific discovery via symbolic regression community, however the clarity of the paper could be improved somewhat.
> >
> > We thank the reviewer for recognizing the contribution and novelty of the proposed SRSD datasets.
> > If neither the revision nor our point-to-point response does not resolve the concern, we would like to know how we can further improve the clarity of this paper.
> >
> > ---
> >
> > > Although the paper proposes a potentially valuable collection of SRSD datasets and a new evaluation metric measuring normalized edit distance, the dataset is not well-documented and it is unclear whether the new evaluation metric provides significantly meaningful insights in terms of scientific discovery compared to existing evaluation metrics.
> >
> > We would like to better understand why the reviewer finds our datasets not well-documented. As explained above, we followed [Hugging Face Dataset Card Creation Guide](https://github.com/huggingface/datasets/blob/main/templates/README_guide.md), which is also [aligned with “Datasheets for Datasets”](https://github.com/huggingface/datasets/blob/main/CONTRIBUTING.md#how-to-contribute-to-the-dataset-cards) for documenting the datasets and code repository. Our three dataset repositories (Easy, Medium, Hard) are documented as README.md files and will be nicely visualized by Hugging Face Datasets/GitHub, providing a link to this paper (Sections 3 and A) for more details.
> >
> > For the comparison with existing evaluation metrics, see our response in [R_CRUL-1st_Round-A2].
> >
> >
> > ---
> >
> > We thank the reviewer for their time to review our paper and provide insightful comments despite the tight review schedule. We will post a follow-up comment to share an additional human-based evaluation to address the main concerns of the reviewer.
> >
> > If the reviewer still needs more clarifications in the meantime to increase the scores, please let us know.

---

> > > ### Comment · Reviewer_CRUL · 2022-11-16
> > > **Response**
> > >
> > > I thank the authors for their response addressing many of my concerns, and I have updated my score accordingly.

---

> > > > ### Author Response · Authors · 2022-11-18
> > > > **A follow-up response to Reviewer CRUL**
> > > >
> > > > We are glad to hear that and thank the reviewer for increasing the score.
> > > > If the reviewer needs further clarifications, please share those with us in this forum. We are still eager to have active discussions until the end of Discussion Stage 2 (Dec 12th).

---

> ### Comment · Reviewer_CRUL · 2022-11-08
> **Response**
>
> I thank the authors for their response in clarifying a number of my concerns, and I look forward to seeing their user study results.

---

> > ### Author Response · Authors · 2022-11-09
> > **1st round Author Response to Reviewer CRUL (Follow-up with user study results)**
> >
> > We thank Reviewer CRUL for waiting.
> >
> > As promised below, we performed a user study with approval from an ethics review board to discuss how aligned with human judges the existing SR and new SRSD evaluation metrics are.
> > Specifically, we recruited 23 people from industry and academia who either have doctoral degrees (scientists, professors, engineers) or are doctoral students, and performed a user study with approval from an ethics review board.
> > The recruited people are in diverse research fields such as computer science, mathematics, physics, chemistry, material science, aerospace engineering, engineering, medical nutrition, and computational biology.
> >
> > Given a pair of true and estimate equations for an SRSD problem, the subjects were asked to assess an estimated equation on a discretized 1-to-5 scale, where 1 and 5 indicate “1: Completely different from the true equation” and “5: Equivalent to the true equation” respectively.
> > We chose SRSD problems among the 120 SRSD datasets such that we can obtain from the experimental results in Section 5 at least two different equations estimated by different methods that are best in terms of $R^2$ score and normalized edit distance.
> > There were 24 resulting SRSD problems for the user study.
> >
> > | **Metrics** |   | **Pearson correlation coefficient** |   |            **P-value** |
> > |:-----------:|---|------------------------------------:|--:|-----------------------:|
> > | $R^2$ score |   | $4.66 \times 10^{-3}$               |   | $0.913$                |
> > | NED         |   | $-0.416$                            |   | $1.85 \times 10^{-24}$ |
> >
> > The table above shows Pearson correlation coefficients (PCCs) between the human judges and SR/SRSD evaluation metrics.
> > For normalized edit distance (NED), the Pearson correlation coefficient and p-value were $-0.416$ and $1.85 \times 10^{-24}$ respectively, which show a much stronger and statistically more significant correlation than one for $R^2$ scores.
> > **In other words, results of the user study indicate that normalized edit distance is more aligned with human judges than $R^2$ score, and thus can be a better estimate about how close to the true equations the estimated equations are, in a more human-understandable way.**
> >
> > We also added the user study to the revision as Section G in appendix. We thank the reviewer for suggesting an additional human-based evaluation in the initial review. We believe that the additional human-based evaluations resolve the reviewer’s concerns and further strengthen this work.
> >
> > If the reviewer still needs more clarifications to increase the scores, please let us know.

---

### Official Review · Reviewer_2CZT · 2022-10-20

**Confidence:** 4
**Correctness:** 3
**Technical Novelty And Significance:** 2
**Empirical Novelty And Significance:** 2
**Recommendation:** 5

**Clarity, Quality, Novelty And Reproducibility:**

The paper is clearly written, and its quality is good.

The novelty is very limited (see above).

**Strength And Weaknesses:**

**Strengths**

The discussion of the limitations of current benchmarks is interesting, and the improvements proposed by the authors (e.g. keeping constants constant, introducing physics-informed ranges for the parameters and variables) makes a lot of sense. Baseline evaluations are provided.

**Weaknesses**

The dataset remains very small, and most if it is a new curation/annotation of an existing benchmark (FRSD). This limits the novelty and impact of this paper. To me, the main problem of symbolic regression benchmarks is the small size of the datasets, and their lack of diversity. The paper does not address these questions.

The edit distance is an interesting idea, but needs a lot of improvements before it can be used as a reliable criterion. Two limitations spring to mind.

* Some weighting of the operators is probably needed. Take f(x) = cos(ax+b), g(x)=sin(ax+b), h(x)=cos(ax), k(x)=exp(ax+b). The edit distances are edit(f, g)=1 edit(f, k)=1 and edit(f, h)=2, yet g is equivalent to f, h is a phase shift away from f (i.e. very close), but k has very different mathematical properties from the three others.
* The magnitude of the constants should be taken into account. For small a, sin(ax) is essentially the same as ax, for large a, the functions are very different. Also, suppose you compare functions of the form f(x)+a.g(x), and u(x)+b.v(x). If a<<1 and b<<1, the edit distance between f and u is probably a much better metric than the edit distance between f+ag and u+bv. This problem will appear every time a function is represented as a sum of terms of decreasing magnitude (a very common situation in science).



**Summary Of The Paper:**

The authors propose a new benchmark set for symbolic regression. Starting from the Feynman symbolic regression dataset (FSRD, Udrescu et al), they propose a new dataset of 120 problems, together with rules for sampling variables and parameters. They introduce a new metric for comparing symbolic functions, based on the edit distance between the trees representing the simplified skeletons of the functions. Finally, they provide baseline comparisons between different popular models (and one introduced by the authors).

**Summary Of The Review:**

The dataset proposed is interesting and welcome addition to current benchmarks and test sets. However, it is very small, and mostly amounts to a new curation of an existing dataset. The edit distance is an interesting idea, but would need significant work before it can be used as a measure of accuracy of symbolic regression.

Overall, my feeling is that the contribution is very marginal, and not novel enough to feature in a venue like ICLR.

---

> ### Author Response · Authors · 2022-11-07
> **1st round Author Response to Reviewer 2CZT (1 of 2)**
>
> We thank Reviewer 2CZT for their comments. We address the concerns with our point-to-point responses below.
>
> ---
>
> ### [R_2CZT-1st_Round-A1]
>
> > Strengths
>
> > The discussion of the limitations of current benchmarks is interesting, and the improvements proposed by the authors (e.g. keeping constants constant, introducing physics-informed ranges for the parameters and variables) makes a lot of sense. Baseline evaluations are provided.
>
> We thank the reviewer for the positive comments.
>
> ---
>
> ### [R_2CZT-1st_Round-A2]
>
> > Weaknesses
>
> > The dataset remains very small, and most if it is a new curation/annotation of an existing benchmark (FRSD). This limits the novelty and impact of this paper.
>
> > To me, the main problem of symbolic regression benchmarks is the small size of the datasets, and their lack of diversity. The paper does not address these questions.
>
> We respectfully disagree with the reviewer about the points for the following reasons:
>
> 1. Each of our 120 SRSD datasets consists of a unique equation, sampling distributions, and 10,000 samples (with ratio of 8:1:1 for train:val:test). It means that our SRSD datasets have 1,200,000 samples in total, which should not be small. We clarified it in the revision (See Section A in appendix).
> 2. Carefully curating/annotating each of SR problems in FRSD for SRSD benchmarks requires non-trivial costs as we review every single variable/constant used in each of 120 equations to make the SR problems more suitable for SRSD (physically make more sense), considering their properties. This contribution should be significant and meaningful for the community.
>
> If the reviewer referred to the total number of SRSD problems (120) instead of the total number of samples (1,200,000), we agree with the reviewer that the number may look small in general.
>
> However, the curation/annotation cost for SRSD problems is non-trivial in practice as a new SRSD problem should be 1) a known physical law which should be unique with respect to 2) skeleton equations (symbolic expressions with constant token(s)) and 3) sampling domains, as explained in Section 3.1 (Duplicate equations). i.e. A new SRSD problem should meet 1) and differ from all the 120 SRSD problems in terms of 2) and/or 3).
>
> Similarly, for the lack-of-diversity issue in existing datasets e.g. FRSD, as pointed out in Section 3.1, we agree with the reviewer as the FRSD datasets have many duplicate SR problems due to their oversimplified sampling domains, which we addressed with our SRSD datasets in this study (See Sections 3.1 and 3.2, Tables 1 and S1-S11.).
>
>
> We emphasize that our focus in this study is on symbolic regression for scientific discovery (SRSD), which has not been well discussed in this community and motivates us to open a discussion to rethink datasets and benchmark specifically for SRSD in this study. To rethink datasets and benchmarks for SRSD and discuss/address critical issues in existing datasets/metrics, this paper should be able to play an important role in this community.
>
> ---
>
> (continued below)

---

> > ### Author Response · Authors · 2022-11-07
> > **1st round Author Response to Reviewer 2CZT (2 of 2)**
> >
> > (continued from above)
> >
> > ---
> >
> > ### [R_2CZT-1st_Round-A3]
> >
> > > The edit distance is an interesting idea, but needs a lot of improvements before it can be used as a reliable criterion. Two limitations spring to mind.
> >
> > > Some weighting of the operators is probably needed. Take f(x) = cos(ax+b), g(x)=sin(ax+b), h(x)=cos(ax), k(x)=exp(ax+b). The edit distances are edit(f, g)=1 edit(f, k)=1 and edit(f, h)=2, yet g is equivalent to f, h is a phase shift away from f (i.e. very close), but k has very different mathematical properties from the three others.
> >
> > > The magnitude of the constants should be taken into account. For small a, sin(ax) is essentially the same as ax, for large a, the functions are very different. Also, suppose you compare functions of the form f(x)+a.g(x), and u(x)+b.v(x). If a<<1 and b<<1, the edit distance between f and u is probably a much better metric than the edit distance between f+ag and u+bv. This problem will appear every time a function is represented as a sum of terms of decreasing magnitude (a very common situation in science).
> >
> >
> > We thank the reviewer for finding use of the edit distance as an interesting idea.
> > As for the idea of weighting, it should be very challenging to 1) mathematically quantify the difference between $\exp$ and others $\sin, \cos$ and 2) justify 1) i.e. the choice of the weighting. As mentioned in Section 4.1, machine translation studies also use edit distance as an evaluation metric. For instance, Przybocki et al. (2006) use edit cost = 1 for all the words regardless of their part of speech tags.
> >
> > Mark Przybocki, Gregory Sanders, and Audrey Le. Edit Distance: A Metric for Machine Translation
> > Evaluation.
> >
> > For both the weighting and magnitude of the constants, we believe that those mathematical properties should be assessed as part of existing error-based metrics such as $R^2$ because we designed
> > In addition, as noted in the conclusion (Section 7), we argue that **the normalized edit distance is a metric not to take the place of existing SR metrics but to incorporate such metrics**, thus we think that a single metric does not have to cover all the aspects. Instead, we can assess the performance of SRSD approach from multiple aspects (multiple metrics), and we proposed the normalized edit distance as one of them.
> >
> > As Reviewer CRUL suggested, we are currently performing a user study to further discuss how meaningful edit distance is, compared to existing metrics, which will also address the reviewer’s concerns. Please refer to a follow-up comment to Reviewer CRUL that we will post later.
> >
> > ---
> >
> > ### [R_2CZT-1st_Round-A4]
> >
> > > The novelty is very limited (see above).
> >
> > > The dataset proposed is interesting and welcome addition to current benchmarks and test sets. However, it is very small, and mostly amounts to a new curation of an existing dataset. The edit distance is an interesting idea, but would need significant work before it can be used as a measure of accuracy of symbolic regression.
> >
> > > Overall, my feeling is that the contribution is very marginal, and not novel enough to feature in a venue like ICLR.
> >
> > Please see our answers in [R_2CZT-1st_Round-A2] and [R_2CZT-1st_Round-A3].
> >
> > We believe that this paper makes significant contributions: 1) the 120 new SRSD datasets (1,200,000 samples and 120 unique problems) with more physics-inspired properties and carefully-curated sampling domains, 2) a benchmark study with a new metric for SRSD, NED, which is designed to assess similarity between true and estimated symbolic expressions as interpretability is a key property of SR. Again, the NED is not a silver-bullet metric, but can incorporate existing SR metrics to assess SR methods from multiple aspects.
> >
> > ---
> >
> > We thank the reviewer for their time to review our paper despite the tight schedule. We believe that our response above resolved major concerns of the reviewer.
> >
> > If the reviewer still needs more clarifications to increase the scores, please let us know.

---

### Official Review · Reviewer_V7wv · 2022-10-28

**Confidence:** 3
**Correctness:** 3
**Technical Novelty And Significance:** 2
**Empirical Novelty And Significance:** 2
**Recommendation:** 5

**Clarity, Quality, Novelty And Reproducibility:**

Clarity: good
Novelty: good
Reproducibility: authors provide various items that may allow the work to be reproduced

**Strength And Weaknesses:**

The main contribution of this work lies in the design of
datasets and benchmark of symbolic regression for scientific
discovery. Authors point our some limitations fo existing datasets and
design their owm dataset categorizing subsets of small, medium and
large complexity and develop a new metric based on tree distance
between the true and the predicted equations.

Where is I in the examples of Table 1? (It is mentioned in the caption
of this table)

"we preprocess equations by 1) substituting constant values e.g., π and
Planck constant to the expression..." --> isn't the idea to learn these
expressions from data? Do you mean that you *postprocess* the learned
formulas?

"coefficient values themselves (e.g., value of C1 in Fig. 2) should
not be important" --> I don't get the idea here. I'd think that
constants can differentiate one expression from another when modeling
real systems (e.g., physics, math etc).

"such true equations will not be available in practice" --> but don't
you generate data from these equations in order to create the
datasets? How do you expect to validate learning models if you don't
have the ground truth? Besides, don't Feynman lectures present the
equations?

Do you have any explanation of why ST results in 0% R2 for all
problems? It seems strange.

Other minor comments:
both the metrics --> both metrics
Table 4 show --> Table 4 shows


**Summary Of The Paper:**

This paper presents the design of
datasets and benchmark of symbolic regression for scientific
discovery. Authors point our some limitations for existing datasets and
design their own dataset categorizing subsets of small, medium and
large complexity and develop a new metric based on tree distance
between the true and the predicted equations.

**Summary Of The Review:**

The main contribution of this work lies in the design of
datasets and benchmark of symbolic regression for scientific
discovery. Authors point our some limitations fo existing datasets and
design their owm dataset categorizing subsets of small, medium and
large complexity and develop a new metric based on tree distance
between the true and the predicted equations.

Where is I in the examples of Table 1? (It is mentioned in the caption
of this table)

"we preprocess equations by 1) substituting constant values e.g., π and
Planck constant to the expression..." --> isn't the idea to learn these
expressions from data? Do you mean that you *postprocess* the learned
formulas?

"coefficient values themselves (e.g., value of C1 in Fig. 2) should
not be important" --> I don't get the idea here. I'd think that
constants can differentiate one expression from another when modeling
real systems (e.g., physics, math etc).

"such true equations will not be available in practice" --> but don't
you generate data from these equations in order to create the
datasets? How do you expect to validate learning models if you don't
have the ground truth? Besides, don't Feynman lectures present the
equations?

Do you have any explanation of why ST results in 0% R2 for all
problems? It seems strange.

Other minor comments:
both the metrics --> both metrics
Table 4 show --> Table 4 shows

---

> ### Author Response · Authors · 2022-11-07
> **1st round Author Response to Reviewer V7wv (1 of 2)**
>
> We thank Reviewer V7wv for their comments and recognizing key strengths of our work.
> We provide our point-to-point responses below.
>
> ---
>
> ### [R_V7wv-1st_Round-A1]
>
> > The main contribution of this work lies in the design of datasets and benchmark of symbolic regression for scientific discovery. Authors point our some limitations fo existing datasets and design their owm dataset categorizing subsets of small, medium and large complexity and develop a new metric based on tree distance between the true and the predicted equations.
>
> We thank the authors for recognizing key contributions of our study.
>
> ---
>
> ### [R_V7wv-1st_Round-A2]
>
> > Where is I in the examples of Table 1? (It is mentioned in the caption of this table)
>
> The property “I” will first appear in Table S1 (appendix).
>
> ---
>
> ### [R_V7wv-1st_Round-A3]
>
> > "we preprocess equations by 1) substituting constant values e.g., π and Planck constant to the expression..." --> isn't the idea to learn these expressions from data? Do you mean that you postprocess the learned formulas?
>
> The reviewer is right. Once an SR model produces a learnt equation from given data, we preprocess learnt equations (formulas) before computing the normalized edit distance (NED) as part of the evaluation process.
>
> ---
>
> ### [R_V7wv-1st_Round-A4]
>
> > "coefficient values themselves (e.g., value of C1 in Fig. 2) should not be important" --> I don't get the idea here. I'd think that constants can differentiate one expression from another when modeling real systems (e.g., physics, math etc).
>
> As explained before the quoted text, the proposed SR metric, NED, is designed to capture similarity between estimated and true equations. The similarity in this context is focused on symbolic expressions to preserve an essential SR property, which is interpretability.
>
> The proposed SR metric is different from existing SR metrics such as $R^2$ score. Such metrics use actual constant values to compute the score/error, but improving SR approaches with respect to $R^2$ score does not guarantee to preserve the property.
>
> As noted in the conclusion (Section 7), we argue that **the normalized edit distance is a metric not to take the place of existing SR metrics but to incorporate such metrics** such as $R^2$ score (Eq. 2), which takes into account the actual constant values.
>
> ---
>
>
> ### [R_V7wv-1st_Round-A5]
>
> > "such true equations will not be available in practice" --> but don't you generate data from these equations in order to create the datasets? How do you expect to validate learning models if you don't have the ground truth? Besides, don't Feynman lectures present the equations?
>
> This is a very important question toward real-world SRSD applications, and we are happy to answer it.
> In this study, we generated datasets from true equations (distributions) **for SRSD benchmark purpose to help the community choose SR methods to be applied for discovering a hidden law (equation) from real-world data.**
>
> If an SR method achieves better overall performance (NED and $R^2$ score) for the large and diverse set of SRSD problems (datasets) than other SR methods do, it would be reasonable to assume that the SR method would better explain observed data in real-world problems (where nobody knows the true equations) by interpretable symbolic expressions.
>
> We emphasize that with the existing SR datasets and metrics (as pointed out in Sections 3 and 4), it would be difficult to discuss SRSD (symbolic regression for scientific discovery), and we believe that the key contribution the reviewer recognized above is very important to the community.
>
> ---
>
> ### [R_V7wv-1st_Round-A6]
>
> > Do you have any explanation of why ST results in 0% R2 for all problems? It seems strange.
>
> Symbolic Transformer (ST) achieved the best performance in terms of NED but still needs more elaborations for coefficient estimation (estimating constant values in a learnt skeleton equation) to improve $R^2$-driven accuracy (Table 2).
>
> Note that Symbolic Transformer is a new baseline with respect to the proposed metric, NED, and **the main contribution of this work lies in the datasets and benchmark of SRSD**, thus Symbolic Transformer itself is not a key contribution of this work as explained in Section 5.1 and Section C (appendix).
>
> ---
>
> (continued below)

---

> > ### Author Response · Authors · 2022-11-07
> > **1st round Author Response to Reviewer V7wv (2 of 2)**
> >
> > (continued from above)
> >
> > ---
> >
> > ### [R_V7wv-1st_Round-A7]
> >
> > > Other minor comments: both the metrics --> both metrics Table 4 show --> Table 4 shows
> >
> > We thank the reviewer for spotting the typo. In our revision, we fixed the typo in “Table 4 show”.
> >
> > “both the metrics” should be fine as it refers to “$R^2$-driven accuracy and solution rate” right before the quoted text.
> >
> > ---
> >
> > ### [R_V7wv-1st_Round-A8]
> >
> > > Clarity: good Novelty: good Reproducibility: authors provide various items that may allow the work to be reproduced
> >
> > We thank the reviewer for the positive comments on clarity, novelty, and reproducibility of this work.
> >
> > ---
> >
> >
> > We thank the reviewer for their effort to review our paper despite the tight schedule. We believe that our response above resolved major concerns of the reviewer.
> >
> > If the reviewer still needs more clarifications to increase the scores, please let us know.

---

### Decision · Program_Chairs · 2023-01-20

**Decision:**

Reject

**Justification For Why Not Higher Score:**

Relatively limited contribution with flaws in one of the main contributions.

**Justification For Why Not Lower Score:**

N/A

**Metareview: Summary, Strengths And Weaknesses:**

This paper presents a new benchmark for symbolic regression, including a new task formulation based on problems that have been used in the literature before, and a new proposal for evaluation metric. While it was unfortunate that there was not a back-and-forth between reviewers and authors on openreview, there was a detailed in-person discussion amongst the AC and reviewers about the paper. The main strength of the paper is that there is a need for improved benchmarks in symbolic regression, and this contribution would likely be used by the community. The main weakness is the proposed evaluation metric. Reviewers have concerns about its usefulness and the validation around it, and are unconvinced that it is ready for widespread adoption by the community in its current form. While the authors did respond to these concerns in the author response, the reviewers still had reservations and were e.g. unconvinced by the analogy to machine translation. The user study is a step in a positive direction but was not enough to sway reviewers that this is a reliable metric ready for widespread adoption.

**Summary Of Ac-Reviewer Meeting:**

Reviewers were generally in agreement that the benchmark would likely be useful but there were serious issues with the new metrics. We spent most of the time discussing how to weight the conflicting forces, and ultimately there was a consensus that it's best for the community to wait until the metric has been sorted out in a more convincing way before accepting the paper.

---

> ### Author Response · Authors · 2023-02-08
> **Author response to meta-review (part 1)**
>
> First of all, we thank the reviewers, AC, and SAC for their work on our submission.
> We are writing this message for potential reviewers of our work at different venues.
>
> On Nov 7, we provided point-to-point responses to all the reviewers and addressed all their comments, including the weakness mentioned in the meta-review. [Reviewer CRUL left a short comment to our response](https://openreview.net/forum?id=i2e2wqt0nAI&noteId=exPtg2Q8t-), but the other reviewers have never responded to our comments. Even though we also asked the AC a few times in private comments to help the reviewers engage in the discussion, we did not see any responses/actions from the AC until the meta-review despite two additional reminders sent by the SAC responding to our requests.
>
> ---
>
> > This paper presents a new benchmark for symbolic regression, including a new task formulation based on problems that have been used in the literature before, and a new proposal for evaluation metric. While it was unfortunate that there was not a back-and-forth between reviewers and authors on openreview, there was a detailed in-person discussion amongst the AC and reviewers about the paper. The main strength of the paper is that there is a need for improved benchmarks in symbolic regression, and this contribution would likely be used by the community.
>
> > The main weakness is the proposed evaluation metric. Reviewers have concerns about its usefulness and the validation around it, and are unconvinced that it is ready for widespread adoption by the community in its current form. While the authors did respond to these concerns in the author response, the reviewers still had reservations and were e.g. unconvinced by the analogy to machine translation. The user study is a step in a positive direction but was not enough to sway reviewers that this is a reliable metric ready for widespread adoption.
>
> The meta-review states *”there was a detailed in-person discussion amongst the AC and reviewers about the paper”*, but **there are no justifications/details provided beyond the existing comments from the reviewers, which we addressed by our point-to-point responses**. Thus, we cannot learn from the comments to improve our work.
>
> The meta-review also argues
> - *“unconvinced by the analogy to machine translation.”*
> - *“The user study is a step in a positive direction but was not enough to sway reviewers that this is a reliable metric ready for widespread adoption.”*
>
> However, **neither the individual comments from the reviewers nor the meta-review explains why they found so**. Otherwise, those may be just their opinions and not supported by evidence, which should be unfair to claim those as weaknesses of our work to reject.
> For these reasons, we respectfully disagree with them for the weakness they claimed.
>
> ---
>
> > Reviewers were generally in agreement that the benchmark would likely be useful but there were serious issues with the new metrics. We spent most of the time discussing how to weight the conflicting forces, and ultimately there was a consensus that it's best for the community to wait until the metric has been sorted out in a more convincing way before accepting the paper.
>
> It seems very unclear that how the following statement comes true, *”it's best for the community to wait until the metric has been sorted out in a more convincing way before accepting the paper”*
> This comment does not suggest anything concrete to improve our work.
>
> Moreover, **the ”serious issues with the new metrics” are not described in either the meta-review or review comments below**. If those were some of the review comments, we provided our response to each of the comments from all the reviewers below, and the meta-review should have pointed out which ones are the cases and should have explained why they found “serious issues” were not addressed by our responses. Otherwise, the assessment should be unfair and we cannot learn from the (meta-)reviews to improve our work.
>
> As repeatedly claimed in the paper and our responses to the reviewers, the NED is not a silver-bullet metric, but can incorporate existing SR metrics to assess SR methods from multiple aspects. There are no perfect evaluation metrics that cover all the essential aspects of a given task, and we proposed the use of the NED for SRSD to cover such aspects that the existing SR metrics cannot address e.g., similarity between the ground-truth and predicted equation trees, which our user study showed more aligned with human judges than the existing metric ($R^2$ score).
> To be widely discussed in the community, the community should accept the work first and keep improving it through several studies rather than waiting for a (nearly) perfect solution from a single paper.
>
>
> ---

---

> > ### Author Response · Authors · 2023-02-08
> > **Author response to meta-review (part 2)**
> >
> > > Relatively limited contribution with flaws in one of the main contributions.
> >
> > As explained above, we emphasize that this claim is not either concrete or supported by evidence.
> > We repeat that if the claim refers to some of the review comments, we provided our response to each of the comments from all the reviewers below, and the meta-review should have pointed out which ones are the cases and should have explained why they found “flaws" were not addressed by our responses. Otherwise, the assessment should be unfair and we cannot learn from the (meta-)reviews to improve our work.